# A Moving-Horizon Approximate Branch-and-Reduce Method for Deep Classification Trees

## Abstract

Despite the importance for interpretability, decision trees face severe scalability challenges. Existing global optimal methods are often limited by binary feature selection and shallow tree depths, whereas traditional heuristic approaches frequently sacrifice predictive accuracy. To overcome these limitations, this paper proposes a moving-horizon approximate branch-and-reduce method to train near-optimal deep classification trees on large-scale datasets with continuous features. Built on a hierarchical root-subtree optimization framework, the method solves the root-level problem via branch-and-reduce while approximating the induced subtree problem using greedy heuristics. Although the underlying framework is capable of guaranteeing global optimality, the approximation, which functions as a lookahead rollout in a reinforcement learning context, significantly boosts efficiency for deeper structures. A low-cost moving-horizon strategy is then employed to iteratively refine model accuracy. Extensive numerical results demonstrate that our method exceeds the testing accuracy of existing heuristic baselines while offering significantly greater scalability, in terms of both dataset size and tree depth, than global optimal solvers.

## 1 Introduction

The decision tree (DT) model is a cornerstone of machine learning, lauded for its interpretable, flowchart-like structure that makes it particularly suitable for classification and regression tasks requiring transparency and comprehensibility (Freitas, 2014; Krzywinski & Altman, 2017). Unlike many black-box models, DTs offer a higher degree of trustworthiness, which is critical in advancing AI for scientific research and high-stakes decision-making (Rudin, 2019; 2022). However, learning an optimal decision tree (ODT) has been classified to be $\mathcal{NP}$-hard (Laurent & Rivest, 1976). Traditional algorithms such as `CART` (Breiman et al., 1984), `ID3` (Quinlan, 1986), and `C4.5` (Quinlan, 1993) have been widely used due to their simplicity and efficiency, employing greedy heuristics to recursively partition data from the root to the leaves. Although these approaches generate trees efficiently, they fall significantly short of achieving optimality, especially for deeper trees.

Recent advancements in deterministic optimization techniques for learning ODTs have focused on approaches leveraging mixed-integer programming (MIP), satisfiability (SAT) solvers (Hu et al., 2020; Alòs et al., 2023), and dynamic programming (DP) approaches (Nijssen & Fromont, 2007; Van der Linden et al., 2022; van der Linden et al., 2023). Among these, MIP-based methods (Günlük et al., 2021; Zhu et al., 2020; Aghaei et al., 2024) formulate the training process as a mixed-integer optimization problem (Bertsimas & Dunn, 2017) and have achieved notable success by providing global optimization guarantees (optimality gap). However, efficiency remains a major limitation of these methods. For example, `Quant-BnB` and `RS-OCT` Mazumder et al. (2022); Hua et al. (2022) have demonstrated the ability to generate optimal trees up to depth 3, extending these approaches to deeper trees remains computationally hard. In contrast, some other scalable DP methods, such as `DL8.5` (Aglin et al., 2020) and `MurTree` (Demirović et al., 2022), leverage advanced caching strategies to enable the learning of relatively deeper trees. Despite these advances, most DP- and SAT-based approaches (Huisman et al., 2024; Lin et al., 2020; Avellaneda, 2020) face inherent drawbacks. They often require binarizing continuous features, which increases the number of features and necessitates approximation techniques from `BinOCT` (Verwer & Zhang, 2019) (some methods directly consider encoding

DTs (Shati et al., 2023)). As a result, these methods solve a feature-binarized approximation of the original problem, which causes information loss, and they still struggle to build trees deeper than 5 on large datasets.

Beyond deterministic approaches, several heuristic methods warrant consideration. One notable example is `TAO` (Carreira-Perpiñán & Tavallali, 2018; Carreira-Perpiñán & Zharmagambetov, 2020; Zharmagambetov et al., 2021), which has been reported to yield only modest improvements over `CART` (Zhu & Shoaran, 2021; Mazumder et al., 2022). More recently, `DPDT` (Kohler et al., 2025) has been proposed, which frames tree induction as a Markov Decision Process in conjunction with `CART`. Although `DPDT` has been shown to outperform `CART`, its further accuracy gains come at a substantial computational cost, which significantly limits its optimality. Within practical time budgets, the trees it produces remain far less accurate than global optimal ones. This gap highlights the urgent need for an advanced method that can deliver both high accuracy and scalability for deep decision trees on large-scale datasets.

This paper introduces the Moving-Horizon Approximate Branch-and-Reduce (`MHABR`) algorithm for training deep classification trees on large-scale datasets with continuous features. This method is designed to achieve better optimization quality than heuristic methods (e.g., `CART`) while remaining substantially more scalable than global optimal solvers. Extensive numerical experiments show that `MHABR` can successfully train trees of depth 8 on datasets with up to 60 million samples, achieving stronger optimization quality than the heuristic baselines we evaluate, while scaling to deeper trees and larger datasets than the global methods considered in our experiments. Additionally, we provide an $\alpha$-tuning option to mitigate overfitting, along with an extended variant that further improves solution quality and brings the result closer to the global optimum.

**Contributions: (1)** We propose a hierarchical root-subtree optimization framework for learning DTs, where the splitting parameters at the root node are optimized according to the values of the two induced child-subtree optimization problems. **(2)** Within this framework, we introduce a branch-and-reduce (`BR`) method for the root-search problem. When the child-subtree problems are solved recursively and exactly, the framework guarantees convergence to global optimality. **(3)** To improve efficiency for deep trees, we approximate the child-subtree problems using `CART`. Under this approximate framework, exactness is retained for depth-2 trees because the depth-1 child-subtree problems are solved exactly. **(4)** We introduce a moving-horizon (MH) technique to iteratively refine branch parameters at low cost, enabling the training of deep trees on large-scale datasets.

**Performance:** We evaluate the proposed method on 59 UCI datasets (Dua & Graff, 2017), with sample sizes ranging from 47 to 60,807,600 and tree depths from 2 to 8, and compare it against six baselines.

- **Testing accuracy:** On 51 small datasets (fewer than 10K samples), `MHABR` achieves the highest average testing accuracy among the compared methods at depths 2 and 3, excluding datasets for which `Quant-BnB` does not converge, and exceeds `DL8.5` by 0.99% on average across the reported depths. On the 8 medium- and large-scale datasets, `MHABR` improves average testing accuracy over `DPDT` by 3.01% at depth 8.
- **Scalability:** On 5 medium datasets (10K–1M samples) and 3 large datasets (1M–60M samples), `MHABR` remains computationally feasible at depth 8, whereas the compared global optimal methods do not complete within the specified time budget. Across these datasets, `MHABR` improves average test accuracy over `CART` by 3.66%.
- **Optimality:** `MHABR` guarantees global optimality for depth-2 trees. On the 42 small datasets completed by `Quant-BnB` at depth 3, its average training-accuracy gap to the global optimum is 0.58%.

## 2 Preliminary

This paper focuses on training a DT model for a classification task. The notation primarily follows Mazumder et al. (2022). We address a supervised learning problem involving a given dataset $\{(x_i, y_i)\}_{i=1}^{n}$, where $x_i \in \mathbb{R}^p$ is the feature vector of sample $i$ and $y_i \in \mathcal{Y}$ is its corresponding class label. Here, $[n] := \{1, \cdots, n\}$, $n$ denotes the number of samples, $p$ denotes the number of features, and $\mathcal{Y} := [n_c] = \{1, \ldots, n_c\}$, $n_c$ denotes the number of classes. The features may be numerical, categorical, or binary. Initially, we scale each feature value of the dataset to the range $[0, 1]$.

### 2.1 Notations for Decision Trees

We develop a balanced binary tree formulation with axis-aligned splits and a fixed depth, denoted by $d$. Let $\mathcal{N}_B := \{1, \cdots, 2^d - 1\}$ and $\mathcal{N}_L := \{2^d, \cdots, 2^{d+1} - 1\}$ denote the sets of branch nodes and leaf nodes, respectively. In these trees, each branch node performs a logical test $x_i^a \leq b$, where $x_i^a$ denotes the value of the feature $a$ for the $i$th sample, thereby partitioning the samples into the left (if "yes") and right (if "no") subtrees. The splitting rule at each branch node is defined by two parameters: the feature indices and the corresponding splitting thresholds, collected in the vectors $\mathbf{a}, \mathbf{b}$, respectively. The partitioning operation is repeated at each layer until every sample is assigned to a leaf node. Each leaf node represents a specific region of the input space and holds a prediction, collected in the vector $\mathbf{c}$.

The induction of a decision tree $T : [0,1]^p \to \mathcal{Y}$ for a group of samples indexed by the set $\mathcal{I}$ can be formulated as an optimization problem expressed as

$$T^*(\mathcal{I}) = \arg\min_{T \in \mathcal{T}_d} L(\mathcal{I}, T), \quad \text{and} \quad V_d(\mathcal{I}) := \min_{T \in \mathcal{T}_d} L(\mathcal{I}, T), \tag{1}$$

where $\mathcal{T}_d$ represents the family of DT of depth $d$, $L(\mathcal{I}, T) := \sum_{i \in \mathcal{I}} \ell(y_i, T(x_i))$ denotes the number of misclassified instances associated with tree $T$ over the samples $\mathcal{I}$, and $\ell := \mathcal{Y} \times \mathcal{Y} \to \mathbb{R}$ denotes the loss function. For classification tasks in this paper, the loss function is defined as $\ell(y_i, \hat{y}_i) = \mathbb{1}\{y_i \neq \hat{y}_i\}$. In this formulation, a penalty term $\alpha \cdot \mathcal{C}$ can be introduced to balance the trade-off between training accuracy and model generalization, where $\mathcal{C}$ denotes the complexity of the decision tree model and $\alpha$ serves as the regularization coefficient. However, this paper focuses on optimizing the training process; thus, the penalty term is omitted in this formulation. The term will later be incorporated into numerical experiments to assess its impact on model performance.

For a sample index set $\mathcal{I} \subseteq [n]$ and feature $a$, let $\mathcal{I}^a$ denote the indices in $\mathcal{I}$ ordered nondecreasingly by $x_i^a$, and let $u_1^a < \cdots < u_{n_a}^a$ be the $n_a$ distinct values of feature $a$ among these samples. Define $u_0^a = u_1^a - 1$ and $u_{n_a+1}^a = u_{n_a}^a + 1$, and let $\mathcal{K}^a := \{0, \ldots, n_a\}$ index the $n_a + 1$ candidate partitions induced by feature $a$. For each $k \in \mathcal{K}^a$, define $\beta_k^a := (u_k^a + u_{k+1}^a)/2$. Since all thresholds in $(u_k^a, u_{k+1}^a)$ induce the same partition, it suffices to consider $\mathcal{B}^a := \{\beta_k^a \mid k \in \mathcal{K}^a\}$ and $\mathcal{B} := \bigcup_{a \in [p]} \mathcal{B}^a$. For each fixed feature $a$, define $\zeta(k) := |\{i \in \mathcal{I} \mid x_i^a \leq \beta_k^a\}|$, where the dependence on $a$ is suppressed for notational simplicity. For $0 \leq k_1 \leq k_2 \leq n_a$, we define

$$\mathcal{I}_{[k_1,k_2]}^a := \{i \in \mathcal{I} \mid \beta_{k_1}^a \leq x_i^a \leq \beta_{k_2}^a\}. \tag{2}$$

In particular, $\mathcal{I}_{[0,k]}^a$ and $\mathcal{I}_{[k,n_a]}^a$ are the disjoint sample sets assigned to the left and right children, respectively, by the split $\beta_k^a$.

### 2.2 A Hierarchical Root–Subtree Framework for Decision Tree Training

We propose a hierarchical root-subtree framework for training decision trees based on the above notation. We denote $T := [a, k, T_L, T_R]$, where $a$ is the feature selected at the root node, $k$ denotes the index of the root split among the candidate thresholds for feature $a$ (that is, the threshold is $\beta_k^a$), and $T_L$ and $T_R$ are the left and right subtrees, respectively. The root node typically exerts the most significant influence on the overall loss in a decision tree (Dwyer & Holte, 2007; Shannon & Banks, 1999), as it processes the entire dataset due to its position at the top of the data flow. Consequently, we isolate the parameters of the root node from the remainder of the tree for optimization through a root-level problem (RLP). The optimal loss is subsequently defined as the sum of the losses from the two subtrees: $T_L$ and $T_R$, optimized in the child-subtree problem (CSP). Then, the optimization problem for training a depth-$d$ tree $T$ can be denoted as:

$$V_d(\mathcal{I}) := \min_{\substack{a \in [p],\ k \in \mathcal{K}^a \\ T_L, T_R \in \mathcal{T}_{d-1}}} L(\mathcal{I}, [a, k, T_L, T_R]) = \min_{a \in [p],\ k \in \mathcal{K}^a} \left\{ V_{d-1}(\mathcal{I}_{[0,k]}^a) + V_{d-1}(\mathcal{I}_{[k,n_a]}^a) \right\}, \tag{3}$$

as well as a hierarchical root-subtree form:

$$\min_{a \in [p],\ k \in \mathcal{K}^a} \left\{ \min_{T_L \in \mathcal{T}_{d-1}} L(\mathcal{I}_{[0,k]}^a, T_L) + \min_{T_R \in \mathcal{T}_{d-1}} L(\mathcal{I}_{[k,n_a]}^a, T_R) \right\}, \tag{4}$$

where the samples split to the left and right subtrees are denoted by $\mathcal{I}_{[0,k]}^a$ and $\mathcal{I}_{[k,n_a]}^a$. In addition, we use a similar notation to define the optimal value when $a$ is fixed, or when $a$ and $k$ are both fixed.

$$V_d(\mathcal{I}, a) = \min_{k \in \mathcal{K}^a} V_d(\mathcal{I}, a, k) = \min_{k \in \mathcal{K}^a} \left\{ V_{d-1}(\mathcal{I}_{[0,k]}^a) + V_{d-1}(\mathcal{I}_{[k,n_a]}^a) \right\}. \tag{5}$$

## 3 A Branch-and-Reduce Method for Optimal Decision Trees

The hierarchical framework, which is explored in `BR` method (detailed in Section 3.2), forms the basis of the approach discussed in this section. Here, we present this approach for solving *RLP*, under the key assumption that *CSPs* can be solved exactly and efficiently.

Within the *RLP*, the decision variables are $a$ and $k$. The variable $a \in [p]$ can be readily enumerated given the typically limited number of features. However, the selection of $k \in \mathcal{K}^a$ is more complex, contingent upon the number of unique feature values $n_a$. For continuous features, $n_a$ can be as large as $n$, rendering the search space prohibitively extensive for exhaustive enumeration. To mitigate this challenge, we propose the `BR` method to determine the optimal $k$ in this section.

### 3.1 Upper Bound, Lower Bound, Reduction, and Branching Strategy

The `BR` method involves four primary steps: Upper Bound, Lower Bound, Reduction Strategy, and Branching Strategy. We now present their details in a general form. Consider a node within the `BR` process, represented by the set of potential split indices $\mathcal{K}_i = \{k \in \mathcal{K}^a \mid l \leq k \leq m\}$, where $0 \leq l \leq m \leq n_a$. In the following description, we specifically select the midpoint $\bar{k} = \lceil \frac{l+m}{2} \rceil$ (the midpoint of $\mathcal{K}_i$) for branching.

**Upper Bound** The upper bound $U$ denotes the best loss among previously evaluated splits, serving as a historical record throughout the iterative search within the feasible set. It is updated iteratively after evaluating a selected index $\bar{k}$ by:

$$U \leftarrow \min\{U, \ V_d(\mathcal{I}, a, \bar{k})\}, \quad \text{where } V_d(\mathcal{I}, a, \bar{k}) = V_{d-1}(\mathcal{I}_{[0,\bar{k}]}^a) + V_{d-1}(\mathcal{I}_{[\bar{k},n_a]}^a). \tag{6}$$

**Boundary Analysis** Define $L(\mathcal{I})$ as the optimal loss of the dataset indexed by $\mathcal{I}$ when training a DT of a fixed depth:

$$L(\mathcal{I}) = \min_{T \in \mathcal{T}} L(\mathcal{I}, T). \tag{7}$$

The following lemma bounds the loss function $L$ in Equation (1).

**Lemma 3.1.** *For two sample index sets $\mathcal{I}_1$ and $\mathcal{I}_2$ with $\mathcal{I}_1 \subseteq \mathcal{I}_2$, let $n_1$ and $n_2$ be the element numbers of $\mathcal{I}_1$ and $\mathcal{I}_2$, where $n_2 \geq n_1$. Then*

$$0 \leq L(\mathcal{I}_2) - L(\mathcal{I}_1) \leq n_2 - n_1.$$

*Proof.* From the formulation of Equation (1), we have:

$$L(\mathcal{I}_1) = \min_{T \in \mathcal{T}} \sum_{i \in \mathcal{I}_1} \ell(y_i, T(x_i)) \leq \min_{T \in \mathcal{T}} \sum_{i \in \mathcal{I}_1 \cup \{\mathcal{I}_2 \setminus \mathcal{I}_1\}} \ell(y_i, T(x_i))$$

$$= \min_{T \in \mathcal{T}} \sum_{i \in \mathcal{I}_2} \ell(y_i, T(x_i)) = L(\mathcal{I}_2). \tag{8}$$

So, we have $L(\mathcal{I}_1) \leq L(\mathcal{I}_2)$, which implies that prediction errors monotonically increase with sample numbers. Suppose the optimal tree for $\mathcal{I}_1$ is $T_1^*$, and for $\mathcal{I}_2$ we have

$$L(\mathcal{I}_2) \leq L(\mathcal{I}_2, T_1^*) = \sum_{i \in \mathcal{I}_2} \ell(y_i, T_1^*(x_i))$$

$$= \sum_{i \in \mathcal{I}_1} \ell(y_i, T_1^*(x_i)) + \sum_{i \in \mathcal{I}_2 \setminus \mathcal{I}_1} \ell(y_i, T_1^*(x_i))$$

$$= L(\mathcal{I}_1) + \sum_{i \in \mathcal{I}_2 \setminus \mathcal{I}_1} \ell(y_i, T_1^*(x_i)), \tag{9}$$

where the first inequality is because $L(\mathcal{I}_2)$ is the optimal value. Then we get:

$$L(\mathcal{I}_2) - L(\mathcal{I}_1) \leq \sum_{i \in \mathcal{I}_2 \setminus \mathcal{I}_1} \ell(y_i, T_1^*(x_i)) \leq \sum_{i \in \mathcal{I}_2 \setminus \mathcal{I}_1} \mathbf{1} = n_2 - n_1. \tag{10}$$

Now we have $0 \leq L(\mathcal{I}_2) - L(\mathcal{I}_1) \leq n_2 - n_1$, which bounds the loss function $L$. $\square$

**Lower Bound**   The lower bound is based on a Lipschitz-type condition, which is illustrated by:

**Lemma 3.2.** *For any $\mathcal{I} \subseteq [n]$ and $d \geq 1$, let $k_1$ and $k_2$ be two split indices corresponding to feature a. Then we have $|V_d(\mathcal{I}, a, k_1) - V_d(\mathcal{I}, a, k_2)| \leq |\zeta(k_1) - \zeta(k_2)|$.*

*Proof.* Without loss of generality, suppose $0 \leq k_1 \leq k_2 \leq n_a$, and let $L(\mathcal{I})$ be defined as in Equation (7). From Equation (3), we have

$$V_d(\mathcal{I}, a, k_1) = L(\mathcal{I}_{[0,k_1]}^a) + L(\mathcal{I}_{[k_1,n_a]}^a), \tag{11}$$

$$V_d(\mathcal{I}, a, k_2) = L(\mathcal{I}_{[0,k_2]}^a) + L(\mathcal{I}_{[k_2,n_a]}^a). \tag{12}$$

Subtracting the two expressions gives

$$\begin{aligned} &V_d(\mathcal{I}, a, k_2) - V_d(\mathcal{I}, a, k_1) \\ &= \left\{ L(\mathcal{I}_{[0,k_2]}^a) - L(\mathcal{I}_{[0,k_1]}^a) \right\} - \left\{ L(\mathcal{I}_{[k_1,n_a]}^a) - L(\mathcal{I}_{[k_2,n_a]}^a) \right\}. \end{aligned} \tag{13}$$

Since $\mathcal{I}_{[0,k_1]}^a \subseteq \mathcal{I}_{[0,k_2]}^a$ and $\mathcal{I}_{[k_2,n_a]}^a \subseteq \mathcal{I}_{[k_1,n_a]}^a$, Lemma 3.1 gives

$$0 \leq L(\mathcal{I}_{[0,k_2]}^a) - L(\mathcal{I}_{[0,k_1]}^a) \leq \zeta(k_2) - \zeta(k_1),$$

$$0 \leq L(\mathcal{I}_{[k_1,n_a]}^a) - L(\mathcal{I}_{[k_2,n_a]}^a) \leq \zeta(k_2) - \zeta(k_1).$$

Using the nonnegativity of the second difference in Equation (13), we obtain

$$V_d(\mathcal{I}, a, k_2) - V_d(\mathcal{I}, a, k_1) \leq \zeta(k_2) - \zeta(k_1).$$

Similarly, using the nonnegativity of the first difference,

$$V_d(\mathcal{I}, a, k_2) - V_d(\mathcal{I}, a, k_1) \geq -\big(\zeta(k_2) - \zeta(k_1)\big).$$

Combining these inequalities yields

$$|V_d(\mathcal{I}, a, k_2) - V_d(\mathcal{I}, a, k_1)| \leq \zeta(k_2) - \zeta(k_1) = |\zeta(k_2) - \zeta(k_1)|. \tag{14}$$

The result follows by symmetry. $\square$

From Lemma 3.2, for a given loss of $\bar{k}$, the lower bound of $\mathcal{K}_i$ is

$$LB := \min\{V_d(\mathcal{I}, a, \bar{k}) - |\zeta(\bar{k}) - \zeta(l)|, V_d(\mathcal{I}, a, \bar{k}) - |\zeta(\bar{k}) - \zeta(m)|\} \tag{15}$$

**Reduction Strategy**   Given $V_d(\mathcal{I}, a, \bar{k})$ and an upper bound $U \leq V_d(\mathcal{I}, a, \bar{k})$, define $\delta(\bar{k}) := V_d(\mathcal{I}, a, \bar{k}) - U$ and $\Delta_i := \left\{ k \in \mathcal{K}_i \mid |\zeta(k) - \zeta(\bar{k})| \leq \delta(\bar{k}) \right\}$. The following lemma shows that no candidate in $\Delta_i$ can strictly improve $U$; hence, we update $\mathcal{K}_i \leftarrow \mathcal{K}_i \setminus \Delta_i$.

**Lemma 3.3.** *Under the above definitions, for any $\mathcal{I} \subseteq [n]$ and $d \geq 1$, every $k \in \Delta_i$ satisfies $V_d(\mathcal{I}, a, k) \geq U$. Therefore, removing $\Delta_i$ while retaining the incumbent solution preserves the global optimum.*

*Proof.* Suppose there is a split $k' \in \Delta_i$ such that $V_d(\mathcal{I}, a, k') < U$. By Lemma 3.2, we have

$$|V_d(\mathcal{I}, a, \bar{k}) - V_d(\mathcal{I}, a, k')| \leq |\zeta(\bar{k}) - \zeta(k')| \leq \delta(\bar{k}) = V_d(\mathcal{I}, a, \bar{k}) - U. \tag{16}$$

However, since $V_d(\mathcal{I}, a, k') < U \leq V_d(\mathcal{I}, a, \bar{k})$, we have

$$|V_d(\mathcal{I}, a, \bar{k}) - V_d(\mathcal{I}, a, k')| = V_d(\mathcal{I}, a, \bar{k}) - V_d(\mathcal{I}, a, k') > V_d(\mathcal{I}, a, \bar{k}) - U, \tag{17}$$

which is a contradiction of Equation (16). Therefore, $V_d(\mathcal{I}, a, k) \geq U$ for every $k \in \Delta_i$. Since the incumbent solution with value $U$ is retained, removing $\Delta_i$ preserves the global optimum. $\square$

**Branching Strategy** To facilitate reduction, the branching strategy bisects the candidate split set at its **midpoint** $\bar{k}$, ensuring a balanced and effective reduction (Ryoo & Sahinidis, 1996; Tawarmalani & Sahinidis, 2004). We can divide $\mathcal{K}_i$ by the midpoint $\bar{k}$ into two subsets: $\mathcal{K}_L$ and $\mathcal{K}_R$ after reduction.

### 3.2 The Procedure of Branch-and-Reduce Method

Now, we are ready to introduce the procedure of our `BR` method. This method is designed to learn global optimal depth-$d$ decision trees. The process begins with an initial estimate (e.g., using `CART`) of the upper bound $U$ and the dataset $\mathcal{I}$. For $a \in [p]$, the algorithm sorts the samples by their feature values to obtain $\mathcal{I}^a$ and constructs the candidate split set $\mathcal{B}^a$. The corresponding index set $\mathcal{K}^a$ contains all split candidates $\beta_k^a$ for $k \in \mathcal{K}^a$, which is then passed to the main loop to obtain the optimal $k$.

---

**Algorithm 1** Branch-and-reduce method (`BR`)

---

1: **function** ExactTreeSearch$(\mathcal{I}, d)$:
2:    Initialize $U$ and $T$ by `CART` ;
3:   **for** $a \in [p]$ **do**
4:      Sort $\mathcal{I}$ to obtain $\mathcal{I}^a$, $\mathcal{K}^a$, and set $\mathbb{K} \leftarrow \{\mathcal{K}^a\}$;
5:      $U, T \leftarrow$ ExactSplitSearch$(U, T, \mathcal{I}^a, a, d, \mathbb{K})$;              *# solve for optimal trees*
6: **return:** Optimal loss $U$ and decision tree $T$.

---

7: **function** ExactSplitSearch$(U, T, \mathcal{I}^a, a, d, \mathbb{K})$:
8:   Initialize the iteration index $i = 0$, $LB = 0$;
9:   **while** $\mathbb{K} \neq \emptyset$ **do**
10:     **node selection**
11:      Select $\mathcal{K}_i$ from $\mathbb{K}$, set $\mathbb{K} \leftarrow \mathbb{K} \backslash \{\mathcal{K}_i\}$, and update $i \leftarrow i + 1$;
12:     **upper bound**
13:      Select the midpoint $\bar{k} \in \mathcal{K}_i$, obtain $\mathcal{I}^a_{[0,\bar{k}]}$, $\mathcal{I}^a_{[\bar{k}, n_a]}$;
14:      $\left( V_{d-1}(\mathcal{I}^a_{[0,\bar{k}]}), \quad T_L \right) \leftarrow$ ExactTreeSearch$(\mathcal{I}^a_{[0,\bar{k}]}, d-1)$,
15:      $\left( V_{d-1}(\mathcal{I}^a_{[\bar{k}, n_a]}), T_R \right) \leftarrow$ ExactTreeSearch$(\mathcal{I}^a_{[\bar{k}, n_a]}, d-1)$;
16:      Evaluate loss $V_d(\mathcal{I}, a, \bar{k}) = V_{d-1}(\mathcal{I}^a_{[0,\bar{k}]}) + V_{d-1}(\mathcal{I}^a_{[\bar{k}, n_a]})$;
17:     **if** $V_d(\mathcal{I}, a, \bar{k}) < U$ **then**
18:      $U \leftarrow V_d(\mathcal{I}, a, \bar{k})$, update $T \leftarrow [a, \bar{k}, T_L, T_R]$;
19:     **lower bound**
20:      Calculate $LB$ by Equation (15), and obtain the optimality gap $U - LB$;
21:     **if** $U - LB \leq 0$ **then**
22:      **continue**                                       *# prune the current region*
23:     **reduction**
24:      $\delta(\bar{k}) \leftarrow V_d(\mathcal{I}, a, \bar{k}) - U$, and get $\Delta_i = \{k \in \mathcal{K}_i : |\zeta(k) - \zeta(\bar{k})| \leq \delta(\bar{k})\}$,
25:     **branching**
26:      Divide the $\mathcal{K}_i$ into $\mathcal{K}_L = \{k \in \mathcal{K}_i \setminus \Delta_i : k < \bar{k}\}$, and $\mathcal{K}_R = \{k \in \mathcal{K}_i \setminus \Delta_i : k > \bar{k}\}$;
27:      Update $\mathbb{K} \leftarrow \mathbb{K} \cup \{\mathcal{K}_L\}$, if $\mathcal{K}_L \neq \emptyset$,      $\mathbb{K} \leftarrow \mathbb{K} \cup \{\mathcal{K}_R\}$, if $\mathcal{K}_R \neq \emptyset$;
28: **return:** Optimal loss $U$ and decision tree $T$.

---

The main loop of the `BR` method solves for $V_d(\mathcal{I}, a)$ with fixed $a$. At each iteration, the steps of this loop are as follows: **(i)** A `BR` node containing the set $\mathcal{K}_i$ is selected from the entire feasible region of $k$, a step referred to as `BR` node selection. **(ii)** The midpoint candidate split $\bar{k}$ is then chosen from $\mathcal{K}_i$. The algorithm evaluates $V_d(\mathcal{I}, a, \bar{k})$, and updates the upper bound if the new solution improves upon the current best result. **(iii)** Computing $\delta(\bar{k})$ by the updated upper bound $U$, the reduction strategy narrows down the set $\mathcal{K}_i$. **(iv)** Once the reduced set is determined, branching occurs on the remaining region. While the lower bound from Equation (15) enables the calculation of the optimality gap and termination based on a specified tolerance, to ensure a fair comparison with state-of-the-art methods, the subsequent method employs a complete search strategy with a tolerance of 0. Therefore, we consider terminating the main loop when $\mathbb{K} = \emptyset$. We can easily prove that Algorithm 1 converges to the **global optimum** of Equation (3) based on Lemmas 3.2 and 3.3.

**Theorem 3.4.** *Algorithm 1 converges to the global optimum of Equation* (3).

*Proof.* Since Equation (3) has a hierarchical structure, we prove the result by induction on the tree depth. We begin with the case of a depth-1 tree (including `R-CART`), where the *CSP* corresponds to optimizing a constant fit. To establish the optimality of Algorithm 1 for a depth-1 tree, it suffices to show that the reduction strategy does not exclude any optimal solution, as the algorithm without reduction effectively performs an exhaustive enumeration. Lemma 3.3 demonstrates that no candidate that can strictly improve the incumbent is removed from the reduced set $\Delta$; therefore, pruning preserves the global optimal objective value.

Next, we assume that Algorithm 1 converges to the global optimal solution for a depth-$(d-1)$ tree. For a depth-$d$ tree, each *CSP* subproblem corresponds to optimizing a depth-$(d-1)$ tree, which, by assumption, is solved to global optimality. The *RLP* follows the same proof structure as the depth-1 case. Consequently, by induction, Algorithm 1 converges to the global optimum of Equation (3). □

*Remark* 3.5. The reduction strategy is the main source of efficiency gains in this algorithm, with a trivial subtraction operation. It is similar to the first lower bounding problem proposed by Mazumder et al. (2022) but is generalized to any depth. The other lower bounding problems proposed by Hua et al. (2022); Mazumder et al. (2022) involve solving complicated lower bounding problems, leading to additional computational overhead. In contrast, `BR` requires only the computation of $\delta(\bar{k})$.

## 4    A Moving-Horizon Approximate Method for Deep Trees

In the above hierarchical optimization framework, we assume that each value $V_{d-1}$ in a *CSP* is computed exactly. Under this assumption, global convergence can be guaranteed for trees of any depth. This condition can be satisfied by recursively calling `BR` for *CSPs*, but it results in an exponential increase in computational time. To address this limitation, this section introduces an approximate method that significantly improves computational efficiency while maintaining high accuracy. Despite the approximation, we show that the global optimality for depth-2 trees is still guaranteed.

### 4.1    Child-Subtree Approximation

In tree structures, the influence of individual nodes on classification loss generally decreases with depth, as deeper nodes tend to affect fewer data points. This implies that *CSPs* have less impact on overall accuracy compared to *RLP*. Consequently, we employ the `CART` algorithm to obtain approximate solutions for *CSPs*. This method offers computational efficiency in addressing the *CSPs*, with limited impact on the optimality, but substantially reducing the overall computational demand. Let $W_d(\mathcal{I})$ denote the loss of a depth-$d$ `CART` tree $\hat{T}$ on dataset $\mathcal{I}$. The `CART` splitting rule for each branch node typically involves minimizing the combined loss of the potential children (detailed in Section 9.1). In our implementation, we use the result of this approximate method to update the upper bound, where the approximate problem is denoted as

$$\hat{V}_d(\mathcal{I}) := \min_{a \in [p], k \in \mathcal{K}^a} \hat{V}_d(\mathcal{I}, a, k) = \min_{a \in [p], k \in \mathcal{K}^a} \{W_{d-1}(\mathcal{I}^a_{[0,k]}) + W_{d-1}(\mathcal{I}^a_{[k,n_a]})\}. \tag{18}$$

Compared to `CART`, this formulation performs a broader root-level search under approximate subtree evaluation, leading to improved accuracy, which is essential for the effectiveness of the following MH procedure. Drawing upon the procedural steps of `BR`, we can obtain an *approximate branch-and-reduce (ABR)* method to solve $\hat{V}_d(\mathcal{I})$, detailed in Algorithm 2. Then, we have the following lemma, stating that **ABR is no worse than CART** in this case.

**Lemma 4.1.** *Let $U_{ABR}(\mathcal{I}, d)$ denote the loss returned by Algorithm 2 for a depth-$d$ tree, and let $W_d(\mathcal{I})$ denote the loss of the depth-$d$ `CART` tree (detailed in Section 9.1) used to initialize the algorithm. Then the following statements hold.*

*1. With or without the reduction strategy, $\hat{V}_d(\mathcal{I}) \leq U_{ABR}(\mathcal{I}, d) \leq W_d(\mathcal{I})$.*
*2. If the reduction is disabled, then $U_{ABR}(\mathcal{I}, d) = \hat{V}_d(\mathcal{I})$.*
*3. For $d = 2$, Algorithm 2 converges to the **global optimal tree**, even when the reduction strategy is applied.*

*Proof.* We first establish the relationship between $\widehat{V}_d(\mathcal{I})$ and $W_d(\mathcal{I})$. If the depth-$d$ `CART` tree is nonterminal, let $(a', k')$ denote the root split selected by `CART`. By the recursive definition of `CART`,

$$W_d(\mathcal{I}) = W_{d-1}(\mathcal{I}^{a'}_{[0,k']}) + W_{d-1}(\mathcal{I}^{a'}_{[k',n_{a'}]})$$
$$= \widehat{V}_d(\mathcal{I}, a', k'). \tag{19}$$

Since $(a', k')$ is a feasible candidate in the minimization defining $\widehat{V}_d(\mathcal{I})$,

$$\widehat{V}_d(\mathcal{I}) = \min_{a \in [p],\, k \in \mathcal{K}^a} \widehat{V}_d(\mathcal{I}, a, k) \leq \widehat{V}_d(\mathcal{I}, a', k') = W_d(\mathcal{I}). \tag{20}$$

If `CART` terminates at the root ($|\mathcal{I}| \leq 1$ or $V_0(\mathcal{I}) = 0$), then $W_d(\mathcal{I}) = \widehat{V}_d(\mathcal{I}) = 0$, so the same inequality holds.

Each candidate root split $(a, k)$ evaluated by Algorithm 2 has loss $\widehat{V}_d(\mathcal{I}, a, k) \geq \widehat{V}_d(\mathcal{I})$. Moreover, by equation 20, the initial `CART` incumbent also satisfies $W_d(\mathcal{I}) \geq \widehat{V}_d(\mathcal{I})$. Since the final incumbent is the minimum of the initial `CART` loss and the losses of the evaluated candidates, while the incumbent is updated only when a strictly smaller loss is found, we obtain

$$\widehat{V}_d(\mathcal{I}) \leq U_{\texttt{ABR}}(\mathcal{I}, d) \leq W_d(\mathcal{I}), \tag{21}$$

regardless of whether reduction is enabled. This proves the first statement.

When the reduction strategy is disabled, Algorithm 2 evaluates every candidate root split $(a, k)$, where $a \in [p]$ and $k \in \mathcal{K}^a$. Therefore,

$$U_{\texttt{ABR}}(\mathcal{I}, d) = \min_{a \in [p],\, k \in \mathcal{K}^a} \widehat{V}_d(\mathcal{I}, a, k) = \widehat{V}_d(\mathcal{I}). \tag{22}$$

Combining this equality with the first statement gives $U_{\texttt{ABR}}(\mathcal{I}, d) = \widehat{V}_d(\mathcal{I}) \leq W_d(\mathcal{I})$, which proves the second statement.

For $d = 2$, each child-subtree problem has depth 1. A depth-1 tree contains only one branch node, and `CART` evaluates the candidate splits at that node exactly. Therefore,

$$W_1(\mathcal{I}^a_{[0,k]}) = V_1(\mathcal{I}^a_{[0,k]}) \text{ and } W_1(\mathcal{I}^a_{[k,n_a]}) = V_1(\mathcal{I}^a_{[k,n_a]}). \tag{23}$$

It follows that $\widehat{V}_2(\mathcal{I}, a, k) = V_2(\mathcal{I}, a, k)$ for every candidate root split $(a, k)$. Taking the minimum over all candidate root splits gives $\widehat{V}_2(\mathcal{I}) = V_2(\mathcal{I})$. Consequently, the Lipschitz-type condition in Lemma 3.2 and the safe reduction result in Lemma 3.3 apply directly to Algorithm 2 when $d = 2$. Thus, the reduction strategy preserves the global optimal objective value, and

$$U_{\texttt{ABR}}(\mathcal{I}, 2) = \widehat{V}_2(\mathcal{I}) = V_2(\mathcal{I}). \tag{24}$$

Therefore, Algorithm 2 returns a **global optimal depth-2 tree**, even when reduction is enabled. $\square$

*Remark* 4.2. Lemmas 3.2 and 3.3 are established for the exact value function $V_d$, for which the child-subtree problems are solved to global optimality. Therefore, their safe-pruning guarantee applies directly to Algorithm 1. For $d > 2$, Algorithm 2 replaces the exact child-subtree values $V_{d-1}$ with the `CART` losses $W_{d-1}$ and consequently evaluates the approximate objective $\widehat{V}_d$. We do not assume or prove that $\widehat{V}_d$ satisfies the Lipschitz-type condition $|\widehat{V}_d(I, a, k_1) - \widehat{V}_d(I, a, k_2)| \leq |\zeta(k_1) - \zeta(k_2)|$. Therefore, for $d > 2$, the reduction step in Algorithm 2 should be understood as a heuristic acceleration strategy. In this setting, we do not claim that reduction preserves the minimizer of $\widehat{V}_d$, that the returned solution equals $\widehat{V}_d(I)$, or that the returned tree is global optimal.

Nevertheless, two validity properties remain. First, because Algorithm 2 is initialized by `CART` and updates the incumbent only when a lower-loss solution is found, its returned training loss is no greater than the `CART` initialization. Second, when reduction is disabled, Algorithm 2 performs an exhaustive root-level search and returns the exact minimizer of the approximate objective $\widehat{V}_d$. Thus, the exact bound theory provides a valid reduction rule for Algorithm 1 and for the depth-2 special case of Algorithm 2, while motivating a computationally effective heuristic reduction for deeper approximate trees.

---

**Algorithm 2** Approximate branch-and-reduce method (ABR)

---

1: **function** ApproxTreeSearch($\mathcal{I}, d$):
2:    Initialize $U$ and $T$ by CART($\mathcal{I}, d$);
3:    **for** $a \in [p]$ **do**
4:       Sort $\mathcal{I}$ by feature $a$ to obtain $\mathcal{I}^a$ and $\mathcal{K}^a$, and set $\mathbb{K} \leftarrow \{\mathcal{K}^a\}$;
5:       $U, T \leftarrow$ ApproxSplitSearch($U, T, \mathcal{I}^a, a, d, \mathbb{K}$);          *# solve the root-level problem*
6:    **return:** Approximate loss $U$ and decision tree $T$.

---

7: **function** ApproxSplitSearch($U, T, \mathcal{I}^a, a, d, \mathbb{K}$):
8:    Initialize the iteration index $i = 0$;
9:    **while** $\mathbb{K} \neq \emptyset$ **do**
10:      **node selection**
11:        Select $\mathcal{K}_i$ from $\mathbb{K}$, set $\mathbb{K} \leftarrow \mathbb{K} \backslash \{\mathcal{K}_i\}$, and update $i \leftarrow i + 1$;
12:      **upper bound**
13:        Select the midpoint $\bar{k} \in \mathcal{K}_i$, and obtain $\mathcal{I}^a_{[0,\bar{k}]}$ and $\mathcal{I}^a_{[\bar{k},n_a]}$;
14:        $\left(W_{d-1}(\mathcal{I}^a_{[0,\bar{k}]}), \widehat{T}_L\right) \leftarrow$ CART($\mathcal{I}^a_{[0,\bar{k}]}, d-1$);   $\left(W_{d-1}(\mathcal{I}^a_{[\bar{k},n_a]}), \widehat{T}_R\right) \leftarrow$ CART($\mathcal{I}^a_{[\bar{k},n_a]}, d-1$);
15:        Evaluate loss $\widehat{V}_d(\mathcal{I}, a, \bar{k}) = W_{d-1}(\mathcal{I}^a_{[0,\bar{k}]}) + W_{d-1}(\mathcal{I}^a_{[\bar{k},n_a]})$;
16:        **if** $\widehat{V}_d(\mathcal{I}, a, \bar{k}) < U$ **then**
17:           $U \leftarrow \widehat{V}_d(\mathcal{I}, a, \bar{k})$, and update $T \leftarrow [a, \bar{k}, \widehat{T}_L, \widehat{T}_R]$;
18:      **reduction**
19:        $\delta(\bar{k}) \leftarrow \widehat{V}_d(\mathcal{I}, a, \bar{k}) - U$, and obtain $\Delta_i = \{k \in \mathcal{K}_i : |\zeta(k) - \zeta(\bar{k})| \leq \delta(\bar{k})\}$;
20:      **branching**
21:        Divide the remaining candidates into $\mathcal{K}_L = \{k \in \mathcal{K}_i \backslash \Delta_i : k < \bar{k}\}$ and $\mathcal{K}_R = \{k \in \mathcal{K}_i \backslash \Delta_i : k > \bar{k}\}$;
22:        Update $\mathbb{K} \leftarrow \mathbb{K} \cup \{\mathcal{K}_L\}$ if $\mathcal{K}_L \neq \emptyset$, and $\mathbb{K} \leftarrow \mathbb{K} \cup \{\mathcal{K}_R\}$ if $\mathcal{K}_R \neq \emptyset$;
23:    **return:** Approximate loss $U$ and decision tree $T$.

---

## 4.2 A Reinforcement Learning Perspective

ABR can be regarded as an approximate DP method (Bertsekas, 2024), and can be explained by Reinforcement Learning (RL) theories. Following DPDT (Kohler et al., 2025), we model the DT problem as a Markov Decision Process. At each layer, the state is defined by the *set of samples* present at the nodes of that layer. The action at a given layer corresponds to the *set of splitting rules* applied to all nodes in that layer. The reward measures the *loss descent* achieved by the splits, calculated as the difference between the total loss at the previous layer and that at the current layer.

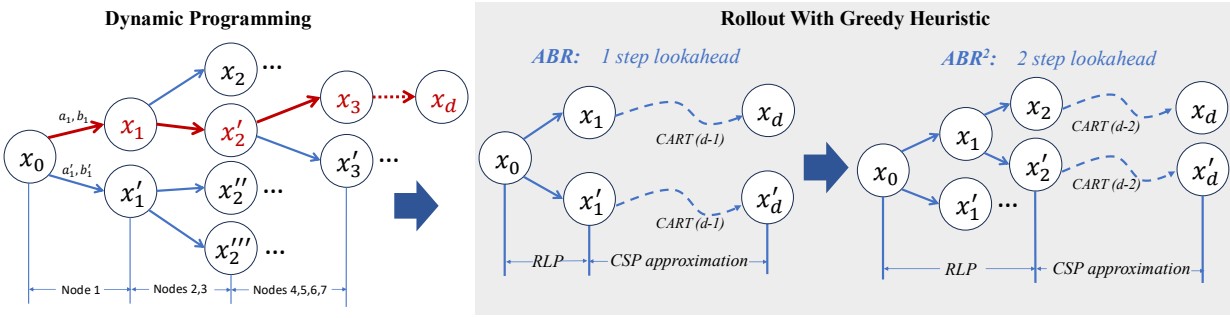

Figure 1: Illustration of the Child-Subtree Approximation via RL ($x$ denotes the state).

For a depth-$d$ tree, the overall objective is to select splitting rules at all layers to minimize the cumulative loss from the root to the leaves. This can be interpreted as a "cost-to-go" problem:

$$1st \; stage \; cost \; + \; optimal \; tail \; problem \; cost.$$

Figure 1 is an illustration of Child-Subtree approximation. Computing the global optimum requires DP that explores all feasible splits, whose number grows exponentially with tree depth. To make this tractable, the *CSP* is approximated in the *value space* (see Section 1.2.3 of Bertsekas (2024)) by constructing a suboptimal policy from two child subtrees using heuristics such as `CART`, replacing the *optimal tail problem cost*. This approximation can be further enhanced by starting the heuristic from stage 2, a 2-step lookahead method, which improves performance at the cost of additional computation.

### 4.3 A Moving-Horizon Approach

The proposed approximation method uses `CART` to generate suboptimal solutions for the left and right subtrees when $d > 2$. To improve solution quality, we introduce the MH approach that iteratively refines branch parameters. At each node, the subsequent nodes form a subtree that is re-optimized with its corresponding samples $\mathcal{I}_{sub}$, creating a new optimization subproblem. As tree depth increases, this iterative refinement helps close the gap between `ABR` and the global optimum. As illustrated in Figure 2, the MH process begins at node 1, where `ABR` updates all branch nodes in the depth-4 tree ($T_{d=4}$, nodes $1, 2, \ldots, 15$). Next, with node 2 as the root, `ABR` refines the branch nodes of the depth-3 subtree ($T_{d=3}$, nodes $2, 4, 5, 8, 9, 10, 11$). This iterative procedure continues in each step, with the subtree depth decreasing until the subtree depth $d_{sub} = 2$. Here, nodes such as $4, 8, 9$ in $T_{d=2}$ are updated three times throughout this process.

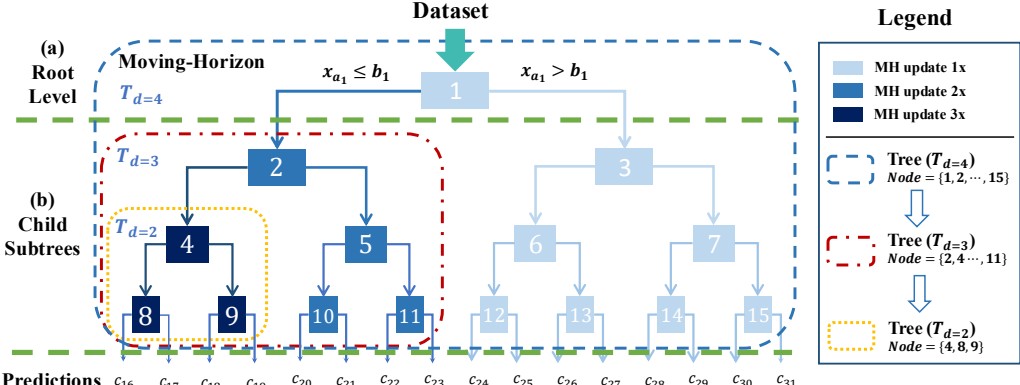

Figure 2: An example of MH when $d = 4$.

After applying the MH refinement at node $t \in \mathcal{N}_B$, we update the corresponding subtree parameters of $T$ using $T_{sub}$ as follows

$$T[t \cdot 2^j : (t+1) \cdot 2^j - 1] \Leftarrow T_{sub}[2^j : 2^{j+1} - 1], \quad j \in \{0, \ldots, d - d_{sub}\}. \tag{25}$$

This iterative refinement can substantially improve the accuracy of deep decision trees, requiring at most $2^{d-1} - 1$ iterations at relatively low cost, and terminates when a subtree reaches an optimal loss, i.e., $d_{sub} = 1$ or $L(\mathcal{I}_{sub}, T_{sub}) = 0$.

### 4.4 Why `ABR` and MH Can Improve Over `CART`

We now give intuition for why the proposed approximation can outperform greedy `CART`. The key difference is that `CART` selects each split using a purely local criterion, whereas `ABR` evaluates a candidate root split by combining it with approximate subtree costs. As a result, `ABR` can prefer a split that appears suboptimal under a one-step greedy criterion but yields a better tree once the downstream subtree structure is considered.

**XOR as an Illustration of Lookahead Benefits** Consider the canonical noiseless XOR dataset (a special case) consisting of the four input patterns $(x^1, x^2) \in \{0, 1\}^2$, each appearing once, with class label $y = x^1 \oplus x^2$. A root split on either $x^1$ or $x^2$ produces two child nodes that each contain one observation from

each class and therefore yields no immediate reduction in classification loss; consequently, a one-step greedy method may fail to identify either feature as informative. In contrast, a depth-2 lookahead recognizes that splitting first on one feature and then on the other perfectly classifies all four observations. This example illustrates how `ABR` can select a root split with little immediate benefit but strong downstream value; the complete loss calculations and resulting tree are provided in Section 9.3.

Although `ABR` uses a stronger root-level objective than `CART`, it still relies on approximate subtree values. The purpose of the MH refinement is to reduce the resulting approximation error by re-optimizing selected subtrees after the higher-level structure has been fixed. In this sense, MH can be viewed as a local repair mechanism: it revisits subproblems that were previously solved only approximately and may further reduce the training loss. For `CART`, the depth-$d$ objective is

$$W_d(\mathcal{I}) = W_{d-1}(\mathcal{I}^{a'}_{[0,k']}) + W_{d-1}(\mathcal{I}^{a'}_{[k',n_{a'}]}), \tag{26}$$

where $(a', k')$ is the greedy root split. In contrast, `ABR` selects the root split according to

$$\hat{V}_d(\mathcal{I}) = W_{d-1}(\mathcal{I}^{a^*}_{[0,k^*]}) + W_{d-1}(\mathcal{I}^{a^*}_{[k^*,n_{a^*}]}), \tag{27}$$

where $(a^*, k^*)$ minimizes the approximate lookahead objective. Thus, `ABR` may select a different root split from `CART`, and the subsequent MH refinements can further improve the resulting tree by re-optimizing selected subtrees. We evaluate this effect empirically in the ablation study of Section 5.6.

### 4.5 A Moving-Horizon Approximate Branch-and-Reduce Method

By leveraging the child-subtree approximation and the MH technique, we present the `MHABR` algorithm for a depth-$d$ tree, as described in Algorithm 3. Designed specifically for trees with $d \geq 2$, the algorithm applies the MH refinement for at most $2^{d-1} - 1$ iterations. At each iteration, it refines the current model by optimizing a selected subtree, using `ABR` (Algorithm 2) to yield a subtree solution $T_{sub}$ with depth $d_{sub}$ and objective value $U_{sub}$. Because `ABR` maintains and updates an incumbent solution over candidate thresholds within the `branch` subroutine, it naturally admits a warm-start strategy. In practice, warm-starting improves efficiency and ensures that the returned subtree solution is no worse than the `CART` initialization. We now analyze monotonicity of the MH refinement and the computational complexity of Algorithm 3.

---

**Algorithm 3** Moving-horizon approximate branch-and-reduce method (`MHABR`)

---

1: **function** `MHABR`$(\mathcal{I}, d)$:
2:     Initialize $U$ and $T$ using `CART`$(\mathcal{I}, d)$;
3:     **for** $t \in \{1, \cdots, 2^{d-1} - 1\}$ **do**
4:         Set the remaining subtree depth as $d_{\text{sub}} \leftarrow d - \lfloor \log_2 t \rfloor$;
5:         Obtain the sample set $\mathcal{I}_t$ reaching node $t$ under the current tree $T$;
6:         **if** $\mathcal{I}_t \neq \emptyset$ **then**
7:             Let $T_t$ denote the current subtree rooted at node $t$, and calculate its loss $U_t \leftarrow L(\mathcal{I}_t, T_t)$;
8:             Calculate $U_{\text{sub}}, T_{\text{sub}} \leftarrow$ `ApproxTreeSearch`$(\mathcal{I}_t, d_{\text{sub}})$;
9:             **if** $U_{\text{sub}} < U_t$ **then**
10:                Update $T$ by replacing $T_t$ with $T_{\text{sub}}$ according to Equation (25);
11:                Update the total loss as $U \leftarrow U - U_t + U_{\text{sub}}$;
12:     **return:** Loss $U$ and decision tree $T$.

---

**Corollary 4.3.** *Let $U^{(r)}$ denote the total training loss after the $r$th accepted moving-horizon update, with $U^{(0)} = W_d(\mathcal{I})$. Then $U^{(r+1)} \leq U^{(r)}$ for every accepted update. Consequently, the final `MHABR` solution satisfies $U_{\text{MHABR}}(\mathcal{I}, d) \leq W_d(\mathcal{I})$, regardless of whether reduction is enabled in the approximate subtree searches.*

*Proof.* At an iteration associated with node $t$, Algorithm 2 replaces the current subtree only if its new loss $U_{\text{sub}}$ is strictly smaller than the current subtree loss $U_t$. The total loss is then updated as $U^{(r+1)} = U^{(r)} - U_t + U_{sub} < U^{(r)}$. If the candidate subtree is not better, no replacement is performed and the total loss remains unchanged. Since the initial tree is obtained by `CART`, $U^{(0)} = W_d(\mathcal{I})$, and hence the final loss cannot exceed $W_d(\mathcal{I})$. $\square$

**Theorem 4.4.** *Let $n > 1$ and $p$ denote the number of samples and features, respectively, and let $\tilde{n} \geq 1$ be the maximum actual number of candidate feature-threshold pairs evaluated in any* `BR` *or* `ABR` *root-level search. Suppose that, whenever a tree or subtree problem containing $n_t \geq 2$ samples is processed, its samples are independently sorted with respect to all $p$ features, requiring $\mathcal{O}(p n_t \log n_t)$ time. Problems containing at most one sample are treated as terminal and are not sorted. We ignore implementation-dependent speedups such as warm-starting, reduction effectiveness, and early termination. Then, for a tree of depth $d$,*

1. *The cost of* `CART` *is bounded by $\mathcal{O}(pnd \log n)$;*
2. *The cost of* `BR` *is bounded by $\mathcal{O}\left(pn \log n \, \tilde{n}^{d-1}\right)$;*
3. *The cost of* `MHABR` *is bounded by $\mathcal{O}\left(\tilde{n} \, pn \log n \, \frac{d(d-1)}{2}\right)$, $d \geq 2$.*

*Since each feature admits at most $n + 1$ candidate thresholds, the total number of candidate feature-threshold pairs satisfies $\tilde{n} \leq p(n + 1) = \mathcal{O}(pn)$. Consequently, substituting $\tilde{n} = \mathcal{O}(pn)$ gives the following coarse worst-case bounds:*

1. *The cost of* `BR` *is bounded by $\mathcal{O}\left((pn)^d \log n\right)$;*
2. *The cost of* `MHABR` *is bounded by $\mathcal{O}\left(p^2 n^2 \log n \, \frac{d(d-1)}{2}\right)$.*

*Proof.* We first establish an inequality that will be used throughout the proof. Consider a collection of mutually disjoint tree nodes with sample sizes $n_1, \ldots, n_r$ satisfying $\sum_{j=1}^{r} n_j \leq n$. For every $n_j > 0$, we have $n_j \leq n$, and hence $\log n_j \leq \log n$. Therefore,

$$\sum_{j=1}^{r} n_j \log n_j \leq \sum_{j=1}^{r} n_j \log n \leq n \log n. \tag{28}$$

**CART**  We consider the `CART` routine used in our implementation, which is based on the `DecisionTree.jl` package, and analyze its complexity under the repeated-sorting assumption of the theorem. Let $C_{\texttt{CART}}(n, d)$ denote the cost of constructing a depth-$d$ `CART` tree from $n$ samples. At a node $t$ containing $n_t$ samples, sorting the samples with respect to all $p$ features costs $\mathcal{O}(p n_t \log n_t)$. After sorting, `CART` scores all candidate thresholds by a linear sweep using cumulative class counts. Since there are at most $p(n_t + 1) = \mathcal{O}(p n_t)$ feature-threshold pairs, this sweep requires $\mathcal{O}(p n_t)$ time and is dominated by the sorting cost.

We prove by induction on $d$ that $C_{\texttt{CART}}(n, d) = \mathcal{O}(pnd \log n)$. For $d = 1$, `CART` processes only the root node. Therefore, $C_{\texttt{CART}}(n, 1) = \mathcal{O}(pn \log n)$. Now suppose that the result holds for trees of depth $d - 1$. Let $n_L$ and $n_R$ denote the numbers of samples assigned to the left and right children of the root, respectively. Since the two child nodes partition the samples, $n_L + n_R = n$. The cost of constructing a depth-$d$ tree satisfies

$$C_{\texttt{CART}}(n, d) \leq \mathcal{O}(pn \log n) + C_{\texttt{CART}}(n_L, d - 1) + C_{\texttt{CART}}(n_R, d - 1)$$
$$\leq \mathcal{O}(pn \log n) + \mathcal{O}(p(d - 1)\left(n_L \log n_L + n_R \log n_R\right)),$$

where the second inequality follows from the induction hypothesis. By Equation (28), we have $n_L \log n_L + n_R \log n_R \leq n \log n$. Hence,

$$C_{\texttt{CART}}(n, d) \leq \mathcal{O}(pn \log n) + \mathcal{O}(pn(d - 1) \log n) = \mathcal{O}(pnd \log n). \tag{29}$$

Thus, the claimed bound holds for `CART`.

**BR**  Let $C_{\texttt{BR}}(n, d)$ denote the cost of solving a depth-$d$ tree exactly using BR. We prove by induction on $d$ that $C_{\texttt{BR}}(n, d) = \mathcal{O}\left(pn \log n \, \tilde{n}^{d-1}\right)$. For $d = 1$, `BR` sorts the samples with respect to all $p$ features and scans the candidate root splits. Therefore, $C_{\texttt{BR}}(n, 1) = \mathcal{O}(pn \log n)$. This agrees with the claimed bound because $\tilde{n}^{d-1} = \tilde{n}^0 = 1$.

Now suppose that the result holds for depth $d - 1$. At the root of a depth-$d$ `BR` problem, at most $\tilde{n}$ candidate splits are evaluated. For candidate split $j$, let $n_{j,L}$ and $n_{j,R}$ denote the sample sizes of the corresponding left

and right subproblems. The two child sample sets partition the parent sample set, so $n_{j,L} + n_{j,R} = n$. The depth-$d$ BR cost therefore satisfies

$$C_{\text{BR}}(n, d) \leq \mathcal{O}(pn \log n) + \sum_{j=1}^{\tilde{n}} \left[ C_{\text{BR}}(n_{j,L}, d-1) + C_{\text{BR}}(n_{j,R}, d-1) \right].$$

Applying the induction hypothesis gives

$$C_{\text{BR}}(n, d) \leq \mathcal{O}(pn \log n) + \mathcal{O} \left( p\tilde{n}^{d-2} \sum_{j=1}^{\tilde{n}} \left[ n_{j,L} \log n_{j,L} + n_{j,R} \log n_{j,R} \right] \right).$$

For each candidate $j$, Equation (28) gives $n_{j,L} \log n_{j,L} + n_{j,R} \log n_{j,R} \leq n \log n$. Since at most $\tilde{n}$ candidates are evaluated,

$$\begin{aligned}
C_{\text{BR}}(n, d) &\leq \mathcal{O}(pn \log n) + \mathcal{O}(p\tilde{n}^{d-2} \cdot \tilde{n}n \log n) \\
&= \mathcal{O}(pn \log n) + \mathcal{O}(pn \log n \, \tilde{n}^{d-1}) \\
&= \mathcal{O} \left( pn \log n \, \tilde{n}^{d-1} \right).
\end{aligned} \tag{30}$$

Thus, the exact BR complexity grows multiplicatively with the number of candidate splits at each recursive depth. Since $\tilde{n} = \mathcal{O}(pn)$,

$$C_{\text{BR}}(n, d) = \mathcal{O} \left( (pn)^d \log n \right). \tag{31}$$

Thus, for fixed $p$ and $n$, the BR bound is exponential in the tree depth $d$; for fixed $d$, it is polynomial in $n$ and $p$, with a degree that increases with $d$.

**One ABR call**  Before introducing MHABR, we consider an ABR call on a subtree containing $n$ samples and having remaining depth $d_{sub} \geq 2$. After initialization, at most $\tilde{n}$ root candidates are evaluated. For candidate $j$, let $n_{j,L}$ and $n_{j,R}$ be the sample sizes of the corresponding left and right child subproblems. The two child-subtree problems induced by each candidate split are approximated using depth-$(d_{sub} - 1)$ CART trees. Using the CART bound,

$$C_{\text{CART}}(n_{j,L}, d_{sub} - 1) + C_{\text{CART}}(n_{j,R}, d_{sub} - 1) \leq \mathcal{O}(p(d_{sub} - 1) \left[ n_{j,L} \log n_{j,L} + n_{j,R} \log n_{j,R} \right]). \tag{32}$$

Because $n_{j,L} + n_{j,R} = n$, by Equation (28), $n_{j,L} \log n_{j,L} + n_{j,R} \log n_{j,R} \leq n \log n$. Hence, the cost of evaluating one candidate root split is bounded by $\mathcal{O} \left( pn(d_{sub} - 1) \log n \right)$. Evaluating at most $\tilde{n}$ root candidates gives

$$\begin{aligned}
C_{\text{ABR}}(n, d_{sub}) &\leq \mathcal{O}(pn \log n) + \sum_{j=1}^{\tilde{n}} \mathcal{O}(pn(d_{sub} - 1) \log n) \\
&= \mathcal{O} \left( \tilde{n} \, pn(d_{sub} - 1) \log n \right).
\end{aligned} \tag{33}$$

The first term accounts for sorting and constructing the candidate root splits. It is dominated by the candidate-evaluation term when $\tilde{n} \geq 1$ and $d_{sub} \geq 2$.

**MHABR**  The moving-horizon procedure applies ABR to subtrees with remaining depths $\{d, d-1, \ldots, 2\}$. Consider one moving-horizon level at which every processed subtree has remaining depth $d_{sub}$. Let these subtrees contain $n_1, \ldots, n_r$ samples. The subtrees processed at a fixed remaining depth $d_{sub}$ are rooted at distinct nodes of the same tree level. Their sample sets are therefore mutually disjoint, and $\sum_{t=1}^{r} n_t \leq n$. By the definition of $\tilde{n}$, each ABR call evaluates at most $\tilde{n}$ root candidates. Applying Equation (33) to each subtree gives

$$\begin{aligned}
\sum_{t=1}^{r} C_{\text{ABR}}(n_t, d_{sub}) &= \mathcal{O} \left( \tilde{n} \, p(d_{sub} - 1) \sum_{t=1}^{r} n_t \log n_t \right) \\
&= \mathcal{O} \left( \tilde{n} \, pn(d_{sub} - 1) \log n \right),
\end{aligned} \tag{34}$$

where the final inequality follows from Equation (28). Summing over all remaining subtree depths $d_{sub} = d, d - 1, \ldots, 2$ gives

$$C_{\texttt{MHABR}}(n, d) = \mathcal{O}\left(\tilde{n}\,pn \log n \sum_{d_{sub}=2}^{d} (d_{sub} - 1)\right) = \mathcal{O}\left(\tilde{n}\,pn \log n \frac{d(d-1)}{2}\right). \tag{35}$$

Finally, since $\tilde{n} = \mathcal{O}(pn)$,

$$C_{\texttt{MHABR}}(n, d) = \mathcal{O}\left(p^2 n^2 \log n \frac{d(d-1)}{2}\right). \tag{36}$$

This completes the proof. □

These bounds are intended only as coarse worst-case estimates; in practice, the observed runtime is substantially reduced by warm-starting, reduction, and early stopping.

## 5 Numerical Experiments

This section presents a comprehensive empirical evaluation of our proposed algorithm across various benchmark datasets (detailed in Section 10). We assess key performance metrics, specifically predictive accuracy and computational efficiency, against five primary baseline methods. These baselines encompass widely used heuristics, including `CART` (Sadeghi et al., 2022), `LS-OCT` (Dunn, 2018), and `DPDT` (Kohler et al., 2025), as well as state-of-the-art global optimization solvers such as `DL8.5` (Aglin et al., 2020) and `Quant-BnB` (Mazumder et al., 2022). Furthermore, due to the lack of an open-source implementation for direct reproducibility, we provide a supplementary comparison against the self-reported results of `TAO` (Carreira-Perpinán & Tavallali, 2018) in Section 8. A central focus of this evaluation is demonstrating three key advantages of our method: (i) it consistently recovers the global optimum at depth $d = 2$; (ii) it achieves vast computational efficiency gains over exact global solvers at depths $d > 2$ with only a marginal trade-off in optimality; and (iii) it can well scale to deep trees while delivering superior accuracy compared to heuristic baselines.

**Datasets and computing environment:** We collected 59 classification datasets from the UCI Machine Learning Repository (Dua & Graff, 2017), spanning both binary and multi-class tasks, with sample sizes ranging from 47 to 60,807,600. The datasets were categorized into 3 groups: 51 small-scale datasets ($n < 10K$) and 5 medium-scale ($10K \le n \le 1M$), and 3 large-scale datasets ($n \ge 1M$). Before the experiments, each dataset was randomly split, with 75% for training and 25% for testing, respectively. The results reflect the average of 10 runs for each dataset. All experiments are conducted on a 40-core *Intel Xeon Gold 5115 CPU* (2.40 `GHz`) with 93.9 `GB RAM`.

**Implementation:** Our algorithm is implemented in `Julia`, with the source code publicly available on GitHub[1]. We also provide an enhanced version of `CART` (Section 9.2), which is based on `DecisionTree.jl` package and is integrated into `MHABR`. All the baselines are obtained from their official repository, except `LS-OCT`, which we reproduced in `Julia`. Time limits are set to 4 hours for small-scale datasets and 24 hours for larger datasets. More details are given in Section 10.3.

### 5.1 Performance on Small Datasets

We evaluate our algorithm on 51 small datasets, encompassing both an aggregate analysis to illustrate the relative performance of the methods and detailed individual comparisons. Since global methods often suffer from limited scalability while heuristic approaches frequently compromise on optimality, we defer a granular analysis of these trade-offs to the scenario-specific discussions.

Table 1 provides overall results of these methods. To rigorously assess optimality, we also utilize binarized datasets to benchmark the optimality gap against `DL8.5`. Among the competing methods, `DPDT` emerges as the strongest baseline; like our approach, it is non-greedy, yet it offers superior efficiency and scalability

---

[1] https://anonymous.4open.science/r/Anonymous_TMLR-B68D

compared to other baselines. Furthermore, `DPDT` can be configured to mimic our root-node search strategy by adjusting the `(N,p)` parameters. Consequently, to ensure a comprehensive comparison, we explicitly evaluate this variant, denoted as `DPDT(Np)`.

Table 1: Average training and testing accuracy on 51 small datasets

| | Depth | Binarized Datasets | | Original Datasets (ODs) | | | | | |
|---|---|---|---|---|---|---|---|---|---|
| | | DL8.5 | MHABR$_{bin}$ | Quant-BnB | LS-OCT | CART | DPDT | DPDT(Np) | MHABR |
| Training accuracy (%) | 2 | 83.18* | 83.18 | **84.31*** | 83.90 | 81.84 | 83.54 | 83.78 | **84.31*** |
| | 3 | 87.71* | 86.97 | 89.44* | 87.59 | 85.96 | 88.19 | 88.02 | 88.93 |
| | 4 | 90.79* | 89.89 | / | 90.14 | 89.02 | 90.95 | 91.00 | **92.26** |
| | 8 | 95.89* | 94.79 | / | 96.94 | 96.70 | 97.72 | 98.01 | **98.55** |
| Testing accuracy (%) | 2 | 79.18 | 75.14 | 79.89 | 79.67 | 78.63 | 79.52 | 79.64 | **79.91** |
| | 3 | 81.56 | 79.82 | 82.56 | 82.18 | 81.13 | 81.96 | 82.08 | **82.42** |
| | 4 | 82.42 | 81.52 | / | **83.53** | 82.67 | 83.30 | 83.16 | 83.30 |
| | 8 | 82.62 | 83.58 | / | 85.20 | 85.20 | **85.35** | 84.95 | 84.10 |

[1] "*" denotes global optimal solutions.    [2] **Bold** numbers indicate the best result.    [3] *bin.* denotes binarized datasets.
[4] Underlined numbers indicate the average over 42 datasets that `Quant-BnB` completes.    [5] `Quant-BnB` fails to converge for depths 4 and 8.

**Accuracy:** As shown in Table 1, `MHABR` achieves the strongest average *testing* accuracy among the compared methods at depths $d = 2$ and $d = 3$ (excluding `Quant-BnB` on datasets for which it does not converge). At $d = 8$, the gap between training and testing accuracy indicates some overfitting on the small datasets, so these results should be interpreted together with validation-based tuning results reported later. Even in this regime, however, `MHABR` maintains an average **0.99%** advantage over `DL8.5`. A broader comparison with non-global baselines on medium- and large-scale datasets, where overfitting is less pronounced, is provided in the following subsection.

In terms of training accuracy, `MHABR` consistently achieves the highest values among the heuristic baselines, outperforming `CART` by an average of **2.63%**. Since training accuracy more directly reflects the quality of the optimized tree on the training objective, we use it here as an indicator of optimization quality rather than generalization performance. Under this metric, `Quant-BnB` is competitive only at $d = 3$, where it completes 42 datasets and yields an average gap of 0.51% relative to `MHABR`, but it fails to converge for deeper trees. On binarized datasets, `MHABR`$_{bin.}$ performs slightly below `DL8.5`, with an average gap of 0.69%, whereas standard `MHABR` on the original continuous datasets exceeds `DL8.5` by an average of 1.62%. Notably, `MHABR` attains the **global optimum** at $d = 2$ on both dataset types. Although `DPDT(Np)` slightly improves upon standard `DPDT` in training accuracy, it remains on average 0.81% below `MHABR`, suggesting that the MH refinements improve optimization quality beyond the one-shot approximate search.

**Scalability and efficiency:** A key property of `MHABR` is its scalability, demonstrated by a slower increase in computational cost with tree depth compared to global optimal methods, while maintaining higher accuracy than heuristic baselines. As shown in Figure 3, across the 51 small-scale datasets, `Quant-BnB` becomes prohibitively expensive for $d \geq 3$, and `DL8.5` exhibits a sharp rise in computational cost with increasing depth, particularly between depths 4 and 8, eventually exceeding the runtime of `MHABR`. In contrast, `MHABR` exhibits nearly linear growth in computational cost, similar to other heuristic methods, which highlights its superior scalability for deeper trees.

## 5.2 Medium-scale and Large-scale Datasets

This section evaluates the proposed method on the challenge of learning deep trees (for $d = 4, 8$) for large-scale datasets, where `Quant-BnB` fails for all cases and `DL8.5` only succeeds on several datasets on depth 4 and fails on depth 8. Table 2 compares the results on five medium and three large datasets against 3 baselines. For large-scale applications of our algorithm, we incorporate additional techniques (detailed in Section 10.2) to improve the efficiency of our algorithm, as detailed in the following, which is further analyzed in Section 5.6.

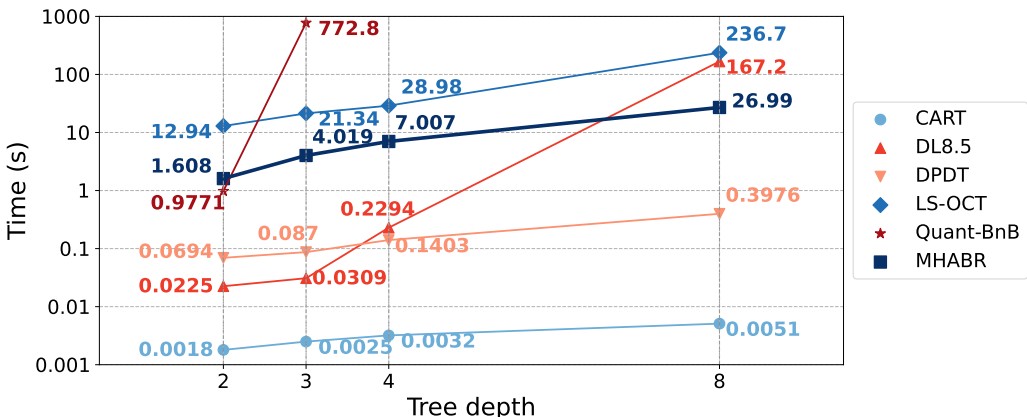

Figure 3: Average training time across 51 small datasets on a logarithmic scale. `CART` and `DPDT` remain the fastest methods, whereas `Quant-BnB` and `DL8.5` exhibit sharp runtime growth with depth. `MHABR` is slower than the heuristic baselines but scales more moderately than the global optimization baselines, requiring 26.99 s at depth 8, compared with 167.2 s for `DL8.5` and 236.7 s for `LS-OCT`.

**Techniques for large-scale datasets: mini-batch and tolerance termination** To enhance efficiency for large-scale datasets, mini-batch sampling and a tolerance termination criterion are employed. Mini-batch sampling uses a subset of the dataset, defined by a sampling ratio $\theta \in (0, 1]$, as input to `ABR`, while the tolerance criterion determines termination. Although this approach may slightly reduce accuracy, it significantly decreases computational time, as demonstrated later. The primary factor affecting computational time is the number of splits in the initial branch, influenced by dataset characteristics, with continuous features often contributing more distinct values, many of which minimally impact the loss function.

For the mini-batch strategy, a proportion $\theta$ of the original data is sampled for each subtree. On large-scale datasets, continuous features often require fewer samples, as their splits generally result in only small fluctuations in the overall loss. To further enhance the efficiency of this algorithm for extremely large-scale datasets, we consider a tolerance parameter $\varepsilon$ as a termination criterion. Let $\texttt{length}(\mathbb{K})$ denote the size of the split index set. The termination criterion can be formally defined as:

$$\texttt{length}(\mathbb{K})/n \leq \varepsilon. \tag{37}$$

For a fixed feature index $a$, the candidate set $\mathcal{K}^a$ of `BR` contains $n_a + 1$ threshold indices. After $j$ successive bisections, any descendant candidate region $\mathbb{K}$ contains at most $\lceil (n_a + 1)/2^j \rceil$ indices. By Equation (37), the search of this region terminates once $\texttt{length}(\mathbb{K}) \leq n \cdot \varepsilon$. Consequently, this condition is reached after at most $\max\{0, \lceil \log_2((n_a + 1)/(n\varepsilon)) \rceil\}$ successive bisections, up to integer rounding.

**Performance on medium and large datasets:** Across the evaluated datasets and baselines, `MHABR` achieves the highest average test accuracy among the compared methods on these datasets, while remaining computationally feasible for deeper trees. On average, it outperforms `CART` by **3.66%** and `DPDT` by **3.01%** in testing accuracy at $d = 8$. On medium datasets, although solvers like `DPDT` often require less runtime, `MHABR` uses additional computation to achieve higher test accuracy on these datasets. Specifically, across the five medium-scale datasets, `MHABR` significantly improves testing accuracy, outperforming `CART` by an average of 1.84% at depth 4 and 4.59% at depth 8, while exceeding `DPDT` by 0.66% at depth 4 and 3.49% at depth 8. The largest improvement is observed on the *Avila* dataset at $d = 8$, where `MHABR` surpasses all other methods by at least 13.35% in training accuracy and 14.53% in testing accuracy. Furthermore, on large-scale datasets where exact methods like `LS-OCT` fail entirely, `MHABR` achieves the highest test accuracy among the compared methods on these datasets. It consistently converges alongside `CART` and `DPDT`, outperforming `CART` by an average of 1.14% at depth 4 and 4.13% at depth 8, while exceeding `DPDT` by 0.63% at depth 4 and 3.89% at depth 8.

Table 2: Performance comparison on 5 medium-scale (top) and 3 large-scale datasets (bottom).

| Dataset | $n$ | $p$ | $n_c$ | Method | $d = 4$ | | | $d = 8$ | | |
|---|---|---|---|---|---|---|---|---|---|---|
| | | | | | Train (%) | Test (%) | Time (s) | Train (%) | Test (%) | Time (s) |
| Avila | 10,430 | 10 | 12 | CART | 57.24 | 56.58 | 0.03 | 76.77 | 74.87 | 0.05 |
| | | | | DPDT | 58.71 | 58.85 | 0.90 | 79.71 | 77.00 | 2.96 |
| | | | | LS-OCT | 60.33 | 59.87 | 183.61 | 79.84 | 77.00 | 1585.04 |
| | | | | MHABR | **61.59** | **60.75** | 8.42 | **93.19** | **91.53** | 14.21 |
| Eeg | 14,980 | 14 | 2 | CART | 70.12 | 69.30 | 0.03 | 79.31 | 76.30 | 0.06 |
| | | | | DPDT | 72.86 | 71.92 | 1.20 | 83.72 | 79.48 | 4.24 |
| | | | | LS-OCT | 71.94 | 71.16 | 326.68 | 81.93 | 78.43 | 2498.00 |
| | | | | MHABR | **74.50** | **72.84** | 23.98 | **88.93** | **82.51** | 102.00 |
| Htru | 17,898 | 8 | 2 | CART | 97.94 | 97.83 | 0.05 | 98.66 | **97.75** | 0.08 |
| | | | | DPDT | 98.14 | **97.92** | 1.63 | 98.87 | 97.68 | 4.82 |
| | | | | LS-OCT | 98.11 | **97.92** | 230.05 | 98.74 | 97.66 | 1897.76 |
| | | | | MHABR | **98.23** | 97.81 | 521.25 | **99.26** | 97.42 | 3363.61 |
| Shuttle | 43,500 | 9 | 7 | CART | 99.83 | 99.80 | 0.03 | **100.00** | 99.95 | 0.04 |
| | | | | DPDT | 99.90 | 99.88 | 0.93 | **100.00** | **99.96** | 1.64 |
| | | | | LS-OCT | 99.92 | 99.89 | 665.52 | **100.00** | **99.96** | 5655.26 |
| | | | | MHABR | **99.97** | **99.95** | 28.53 | **100.00** | 99.95 | 30.09 |
| Skin-seg. | 245,057 | 3 | 2 | CART | 97.42 | 97.37 | 0.08 | 99.50 | 99.49 | 0.10 |
| | | | | DPDT | 98.22 | 98.20 | 5.48 | 99.76 | 99.74 | 10.58 |
| | | | | LS-OCT | 98.41 | 98.39 | 1471.57 | 99.75 | 99.73 | 15259.16 |
| | | | | MHABR | **98.72** | **98.72** | 46.62 | **99.95** | **99.91** | 164.83 |
| SUSY | 5,000,000 | 18 | 2 | CART | 76.60 | 76.61 | 29.32 | 78.38 | 78.34 | 75.79 |
| | | | | DPDT | 76.86 | 76.87 | 1197.44 | 78.72 | 78.68 | 2951.08 |
| | | | | MHABR | **77.84** | **77.86** | 17632.26 | **78.94** | **78.89** | 81290.06 |
| HIGGS | 11,000,000 | 28 | 2 | CART | 65.58 | 65.55 | 81.98 | 69.47 | 69.40 | 216.10 |
| | | | | DPDT | 66.20 | 66.18 | 3223.50 | 69.67 | 69.57 | 7158.52 |
| | | | | MHABR | **66.88** | **66.85** | 11826.94 | **70.17** | **70.05** | 78370.11 |
| WESAD | 60,807,600 | 8 | 8 | CART | 57.46 | 57.46 | 123.23 | 80.27 | 80.26 | 293.24 |
| | | | | DPDT | 61.21 | 61.20 | 5225.47 | 79.45 | 79.45 | 14623.90 |
| | | | | MHABR | **61.33** | **61.33** | 1464.82 | **85.37** | **85.38** | 7977.35 |

[1] **Bold** numbers indicate the best result among all the methods.  [2] LS-OCT fails to converge on large datasets.
[3] MHABR is a light version of MHABR incorporating additional techniques.

**Runtime Determinants**  The training time of MHABR is not necessarily monotone in the sample size $n$, because its dominant computational burden is determined more directly by the number of unique split candidates, characterized by $n_a$ and $|\mathcal{K}^a|$. As shown in Table 2, Shuttle contains substantially more observations than Htru, yet requires considerably less training time. A similar non-monotone relationship is also observed for Quant-BnB; for example, Skin is larger than Avila but is solved in less time (Mazumder et al., 2022). These observations indicate that sample size alone is an incomplete predictor of computational cost. Repeated feature values do not create additional candidate thresholds, while dataset-dependent reduction and early termination can further contract the effective search space.

## 5.3  Compared to DPDT

DPDT is one of the strongest heuristic baselines in our experiments. We therefore provide a more detailed comparison in this subsection. We evaluate DPDT under different parameter configurations and with $\alpha$-tuning to show that MHABR achieves better optimization quality while remaining computationally efficient under a comparable search space.

Recall the training results of $d = 8$ across 51 datasets in Table 1; more specifically, `MHABR` strictly outperforms `DPDT` on 23 datasets with an overall improvement of 46.29%, while being only 3.93% worse on 7 datasets. These results demonstrate that `MHABR` achieves better optimization quality compared to default `DPDT`.

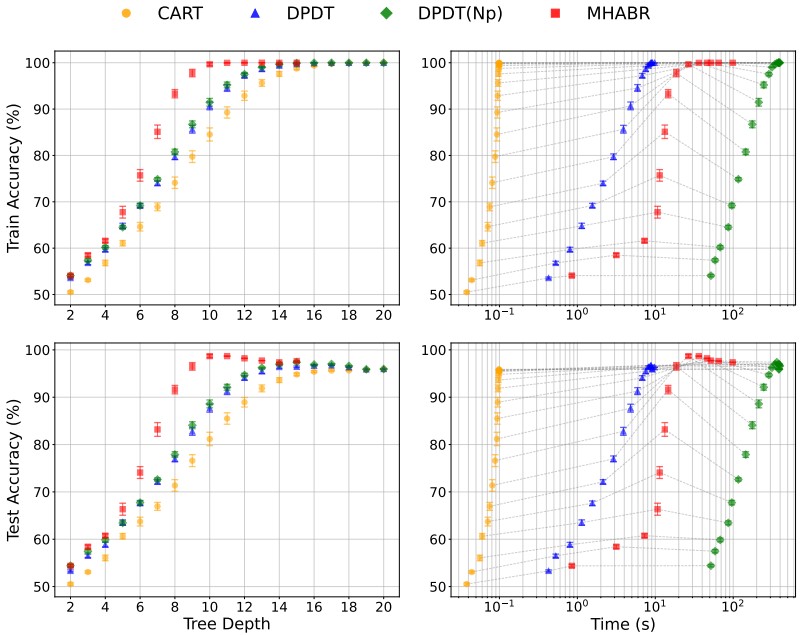

Figure 4: Training/testing accuracy vs. tree depth/running time on `Avila`

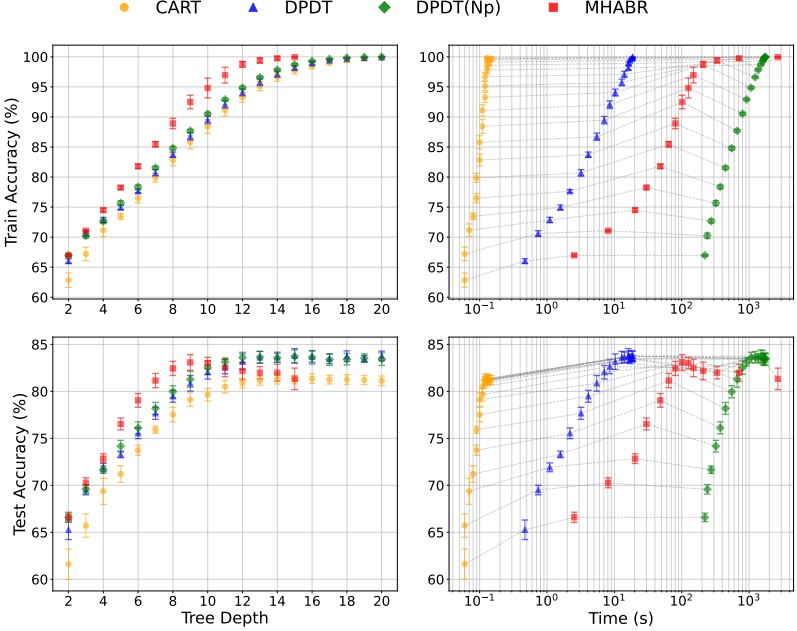

Figure 5: Training/testing accuracy vs. tree depth/running time on `Eeg`

For a detailed comparison with `DPDT` under comparable settings, we also compare to `DPDT(NP)`, `cart_nodes_list=(Np, )`, which enables it to achieve the same theoretical accuracy as `ABR`. To illustrate the performance gains brought by the reduction and MH techniques, we use the datasets `Avila` and `Eeg` to plot the corresponding Pareto fronts for `CART`, `DPDT`, `DPDT(NP)`, and `MHABR`. We evaluate tree depths from 2

to 20. For `MHABR`, we report results only up to $d = 15$, since it already reaches almost 100% training accuracy at depth 10 and begins to show signs of overfitting thereafter.

The Pareto fronts shown in Figures 4 and 5 illustrate the trade-offs between training/testing accuracy and tree depth, as well as between training/testing accuracy and training time. To clearly show the relationship between tree depth and running time, we also connect the points of each same depth in the figures of training/testing accuracy vs. time. In Figure 4, among all methods, `MHABR` achieves the highest training accuracy and the highest peak test accuracy among the compared methods in this experiment. It achieves almost perfect training accuracy 99.66% at depth $d = 10$, while its testing accuracy also achieves the highest peak performance 98.68%. Beyond this regime, its testing accuracy declines, reflecting its tendency to overfit at larger depths. In contrast, `DPDT` and `DPDT(Np)` approach this level only at depths $d \geq 14$, whereas `CART` approaches it only at depths $d \geq 17$. In Figure 5, `MHABR` shows weaker testing accuracy when $d \geq 10$, which is expected due to overfitting, as its training accuracy is nearly 100%. This leads to a drop in testing accuracy while running time increases. Nonetheless, excluding overfitting, `MHABR` still outperforms the other methods.

In terms of running time, `CART` is the fastest method, but it requires substantially deeper trees (i.e., much larger models) to achieve competitive accuracy. `DPDT` improves accuracy relative to `CART` with only a modest increase in computation, but its strength is mainly evident in its default configuration. For achieving higher accuracy, `MHABR` performs better and requires less time than `DPDT(Np)`. Since model size is crucial for the interpretability of decision trees, obtaining high accuracy with a smaller model is particularly desirable, which is an advantage offered by `MHABR`.

`MHABR` achieves higher accuracy than `DPDT(Np)` at the same depth (before overfitting), demonstrating the substantial performance gains contributed by the MH component. Moreover, `MHABR` requires less training time than `DPDT(Np)` before overfitting occurs, highlighting its superior efficiency enabled by the Reduction technique. Although the slope of the running-time increase for `MHABR` becomes steeper than that of `DPDT(Np)` once the depth exceeds 13, an effect attributable to the MH procedure, consistent with our complexity analysis, the training accuracy has already reached 100% at this point and cannot improve further. Thus, MH iterations can be relaxed or omitted beyond this depth.

Table 3: $\alpha$-tuning results of 25 datasets with $n \geq 500$.

| Depth | Training Accuracy | | | | Testing Accuracy | | | |
|---|---|---|---|---|---|---|---|---|
| | CART | DPDT | DPDT(Np) | MHABR | CART | DPDT | DPDT(Np) | MHABR |
| **2** | 81.29 (1.19) | 82.54 (1.13) | 82.57 (1.06) | **82.89** (1.03) | 80.07 (1.94) | 81.05 (1.87) | 81.09 (1.93) | **81.34** (3.16) |
| **3** | 84.29 (1.49) | 85.69 (1.33) | 85.78 (1.19) | **86.54** (1.12) | 82.57 (1.85) | 83.47 (1.61) | 83.54 (1.71) | **83.92** (1.77) |
| **4** | 86.66 (1.44) | 87.93 (1.56) | 88.23 (1.23) | **89.20** (1.53) | 84.43 (1.79) | 84.87 (1.69) | 84.77 (1.76) | **85.40** (1.90) |
| **5** | 88.35 (1.91) | 90.27 (2.12) | 90.47 (1.90) | **91.76** (1.77) | 85.42 (2.02) | 86.11 (2.05) | 86.26 (2.12) | **86.31** (1.95) |

$\alpha$-**tuning** To evaluate the methods under a realistic model-selection setting, we split the data into 50% for training, 25% for validation, and 25% for testing subsets. For every method, $\alpha$ is selected from $\alpha \in \{0.0, 0.001, 0.005, 0.01, 0.05, 0.1, 0.2\}$, using the same validation-based tuning procedure. We consider 25 datasets with more than 500 samples and evaluate tree depths from 2 to 5, thereby reducing the instability associated with validation-based selection on very small datasets. As shown in Table 3, `MHABR` achieves the highest average training and testing accuracy at every evaluated depth under this common tuning protocol. In particular, its consistent advantage in testing accuracy indicates that `MHABR` provides the strongest overall predictive performance among the compared methods after $\alpha$-tuning. Because $\alpha$ is selected by validation, Table 3 evaluates model-selection and predictive performance, not the ability of each method to minimize a common unpenalized training objective.

## 5.4 Discussion about comparison with global optimal methods

In this subsection, we provide a comparative analysis of `MHABR` and global optimal methods: `DL8.5` and `Quant-BnB`, focusing on small datasets for which all evaluated methods successfully identified feasible solutions.

The results show that our method achieves higher training accuracy than `DL8.5` in this setting and offers a favorable empirical trade-off between optimization quality and runtime relative to `Quant-BnB`.

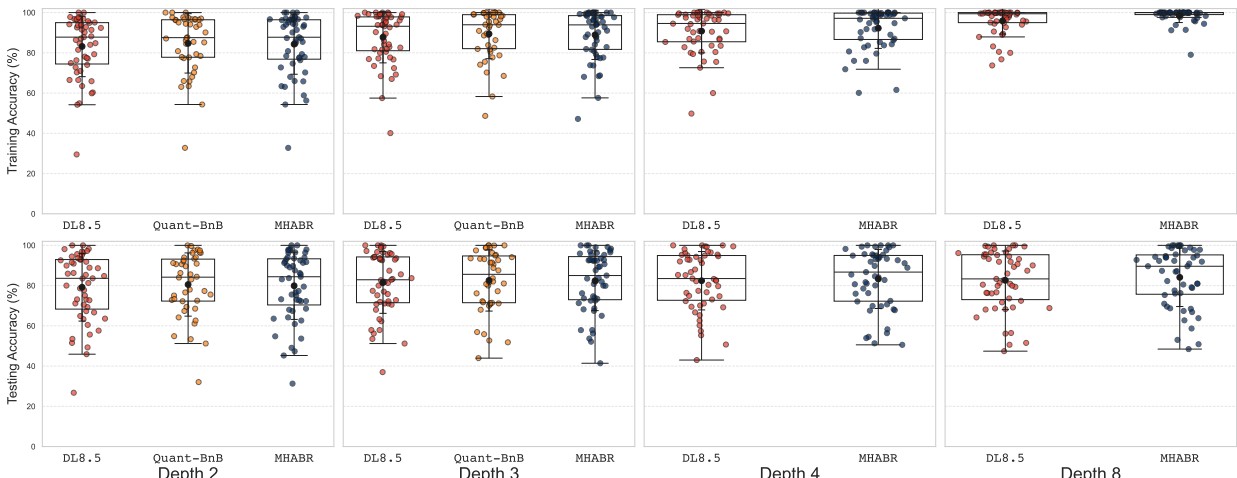

Figure 6: Training and testing accuracy of 51 small datasets across $d = 2, 3, 4, 8$.

We use box plots, as shown in Figure 6, to illustrate the results across all datasets. Although `DL8.5` is a global optimal method, it relies on the binarization of continuous datasets. This preprocessing step introduces approximation errors, leading to worse training accuracy compared to the other two methods. This negative impact is particularly evident in the training results at depth 8. In contrast, `MHABR` achieves higher training accuracy than `DL8.5` on these datasets, especially at depth 8. Meanwhile, `Quant-BnB` can handle the original continuous datasets natively but is limited to a maximum depth of 3 (successfully completing only 42 datasets). Table 4 presents the results for these 42 datasets at depths 2 and 3. `MHABR` achieves training accuracy equivalent to that of `Quant-BnB` at $d = 2$. At $d = 3$, it demonstrates only a minor disparity, performing marginally lower than `Quant-BnB`. This outcome suggests that the solutions generated by `MHABR` closely approximate the global optimal solutions for shallow trees. Furthermore, it indicates that the potential loss of optimality resulting from the *CSP* approximation is effectively compensated for by the MH procedure.

Table 4: Comparison on 42 small datasets (excluding 9 datasets on which `Quant-BnB` fails to complete).

|          | Depth   | CART              | DL8.5                | Quant-BnB              | MHABR               |
|----------|---------|-------------------|----------------------|------------------------|---------------------|
| Train (%) | $d = 2$ | $84.10 \pm 14.11$ | $83.93 \pm 15.04^*$  | $\mathbf{84.70 \pm 14.76}^*$ | $\mathbf{84.70 \pm 14.76}^*$ |
|          | $d = 3$ | $87.99 \pm 12.41$ | $88.08 \pm 13.18^*$  | $\mathbf{89.44 \pm 12.36}^*$ | $88.86 \pm 12.62$   |
| Test (%)  | $d = 2$ | $78.98 \pm 16.37$ | $79.89 \pm 16.40$    | $\mathbf{80.53 \pm 15.71}$   | $80.47 \pm 15.91$   |
|          | $d = 3$ | $81.41 \pm 14.97$ | $81.71 \pm 15.79$    | $\mathbf{82.56 \pm 15.22}$   | $82.30 \pm 15.14$   |

[1] **Bold** numbers indicate the best solutions.    [2] "$*$" signifies the global optimum in the corresponding formulation.
[3] Even though `Quant-BnB` and `MHABR` can obtain the optimal loss when $d = 2$, the optimal trees might differ.

## 5.5 Optimality Gap Evaluation and An Extension

To assess the optimality of our algorithm, we evaluate 10 datasets (from the 51 small datasets) and compute the optimality gap as the relative difference between the solution produced by `MHABR` and the global optimum obtained by `BR`. We further introduce an extended variant, `MHABR`[2] (2-step lookahead method), which reduces the optimality gap by extending the *RLP* from the root node to the top two layers (nodes $\{1, 2, 3\}$) for $d \geq 3$. Importantly, `MHABR`[2] attains exact optimal solutions for depth-3 trees and further reduces the optimality gap for deeper trees.

`MHABR`[2]: Our algorithm is implemented in `Julia`. For the default version, we adopt the reduction strategy with parameters $\varepsilon = 0$ and $\theta = 1$. To construct `MHABR`[2], which enlarges the *RLP* to include the top two layers

of nodes, we modify the evaluation function in `ABR` recursion as follows:

$$\texttt{MHABR}: \begin{cases} \texttt{BR}, & d - d_{sub} \leq 1 \\ \texttt{CART}, & \text{otherwise} \end{cases} \Rightarrow \texttt{MHABR}^2: \begin{cases} \texttt{BR}, & d - d_{sub} \leq 2 \\ \texttt{CART}, & \text{otherwise}. \end{cases}$$

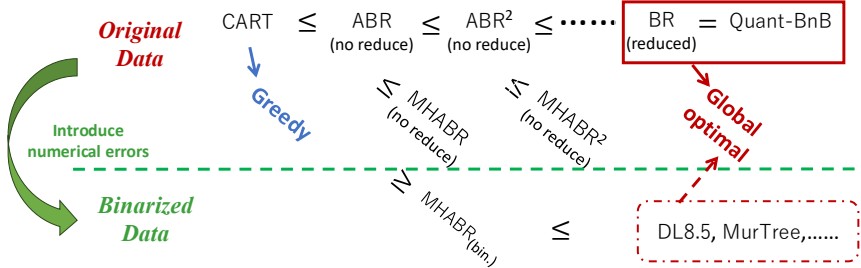

Figure 7: Theoretical relationships in terms of optimality between our algorithms and other global optimal methods

First, we illustrate the scope of our method in Figure 7, which summarizes the theoretical relationships among the proposed approaches. As noted in Lemma 4.1, our algorithm guarantees solutions no worse than `CART` when no reductions are applied, with the MH procedure further refining results toward better optimality. Expanding the `RLP` scope improves the approximation, making `ABR`$^2$ superior to `ABR`, and further enlargement brings methods closer to global optimal approaches such as the unapproximated `BR` and `Quant-BnB`. For binarized datasets, the binarization process introduces errors that reduce optimality, so `MHABR`$_\text{bin.}$ performs worse than the original dataset version and is clearly inferior to global optimal methods such as `DL8.5` and `MurTree`.

We evaluate the optimality gap using 10 small datasets solvable by `BR`, with results reported in Tables 5 and 6. As shown, `CART` exhibits a relatively large gap, typically exceeding 3% on average at both depths 3 and 4. In contrast, `MHABR` maintains a gap within 1% on average, while the extended variant, `MHABR`$^2$, further improves optimality, consistently reducing the gap below 1%. Correspondingly, the computational cost increases from `CART` to `BR` as optimality improves. Therefore, we generally recommend `MHABR`, which offers strong optimality while keeping the computational cost reasonable.

Table 5: Comparison of `CART`, `MHABR`, `MHABR`$^2$, and `BR` against the global optimum across 10 datasets ($d = 3$)

| Dataset | $n$ | $p$ | $n_c$ | CART (greedy) | | MHABR | | MHABR$^2$ (optimal) | | BR (optimal) | |
|---|---|---|---|---|---|---|---|---|---|---|---|
| | | | | Gap (%) | Time ($s$) | Gap (%) | Time ($s$) | Gap (%) | Time ($s$) | Gap (%) | Time ($s$) |
| Iris | 150 | 4 | 3 | 2.68 | 0.0004 | 0.18 | 0.02 | 0.00 | 0.09 | 0.00 | 0.09 |
| Haberman-survival | 306 | 3 | 2 | 3.97 | 0.0002 | 0.69 | 0.01 | 0.00 | 0.12 | 0.00 | 0.12 |
| Monks-problems-3 | 554 | 6 | 2 | 2.41 | 0.0002 | 2.46 | 0.01 | 0.00 | 0.02 | 0.00 | 0.02 |
| Monks-problems-1 | 556 | 6 | 2 | 12.15 | 0.0005 | 1.02 | 0.01 | 0.00 | 0.02 | 0.00 | 0.02 |
| Monks-problems-2 | 600 | 6 | 2 | 2.70 | 0.0019 | 0.45 | 0.01 | 0.00 | 0.02 | 0.00 | 0.02 |
| Balance-scale | 625 | 4 | 3 | 3.45 | 0.0001 | 1.09 | 0.01 | 0.00 | 0.02 | 0.00 | 0.02 |
| Blood-transfusion | 748 | 4 | 2 | 2.36 | 0.0001 | 1.03 | 0.02 | 0.00 | 0.45 | 0.00 | 0.45 |
| Mammographic-mass | 830 | 5 | 2 | 1.16 | 0.0004 | 0.22 | 0.02 | 0.00 | 0.31 | 0.00 | 0.31 |
| Contraceptive-method-choice | 1,473 | 9 | 3 | 9.95 | 0.0010 | 1.00 | 0.03 | 0.00 | 0.51 | 0.00 | 0.51 |
| Car-evaluation | 1,728 | 6 | 4 | 3.72 | 0.0001 | 0.60 | 0.01 | 0.00 | 0.07 | 0.00 | 0.07 |

## 5.6 Ablation Studies

We now study how the reduction strategy, the MH method, and tunable parameters $\varepsilon$ and $\theta$ affect the computational efficiency and accuracy of `MHABR`, by systematically evaluating MH procedure performance per layer using an `Avila` case study on a depth-8 decision tree.

To evaluate the contribution of the MH refinement, we show the results of a depth-8 tree on the Avila dataset, without any other techniques except the approximation. The result is shown in Figure 8(a). We observe that

Table 6: Comparison of `CART`, `MHABR`, `MHABR`$^2$, and `BR` against the global optimum across 10 datasets ($d = 4$)

| Dataset | $n$ | $p$ | $n_c$ | CART (greedy) | | MHABR | | MHABR$^2$ | | BR (optimal) | |
|---|---|---|---|---|---|---|---|---|---|---|---|
| | | | | Gap (%) | Time ($s$) | Gap (%) | Time ($s$) | Gap (%) | Time ($s$) | Gap (%) | Time ($s$) |
| Iris | 150 | 4 | 3 | 0.54 | 0.0005 | 0.00 | 0.02 | 0.00 | 0.30 | 0.00 | 8.36 |
| Haberman-survival | 306 | 3 | 2 | 6.94 | 0.0002 | 1.71 | 0.03 | 0.20 | 0.24 | 0.00 | 5.05 |
| Monks-problems-3 | 554 | 6 | 2 | 0.00 | 0.0002 | 0.00 | 0.02 | 0.00 | 0.04 | 0.00 | 0.24 |
| Monks-problems-1 | 556 | 6 | 2 | 16.26 | 0.0006 | 2.48 | 0.02 | 0.07 | 0.04 | 0.00 | 0.25 |
| Monks-problems-2 | 600 | 6 | 2 | 4.46 | 0.0021 | 1.13 | 0.03 | 0.64 | 0.05 | 0.00 | 0.27 |
| Balance-scale | 625 | 4 | 3 | 5.66 | 0.0001 | 0.78 | 0.03 | 0.40 | 0.06 | 0.00 | 0.39 |
| Blood-transfusion | 748 | 4 | 2 | 4.00 | 0.0001 | 1.23 | 0.08 | 0.49 | 1.08 | 0.00 | 34.71 |
| Mammographic-mass | 830 | 5 | 2 | 2.36 | 0.0005 | 0.62 | 0.07 | 0.22 | 0.73 | 0.00 | 15.38 |
| Contraceptive-method-choice | 1,473 | 9 | 3 | 5.76 | 0.0011 | 1.19 | 0.11 | 0.27 | 1.40 | 0.00 | 28.79 |
| Car-evaluation | 1,728 | 6 | 4 | 3.82 | 0.0001 | 0.11 | 0.05 | 0.00 | 0.15 | 0.00 | 1.24 |

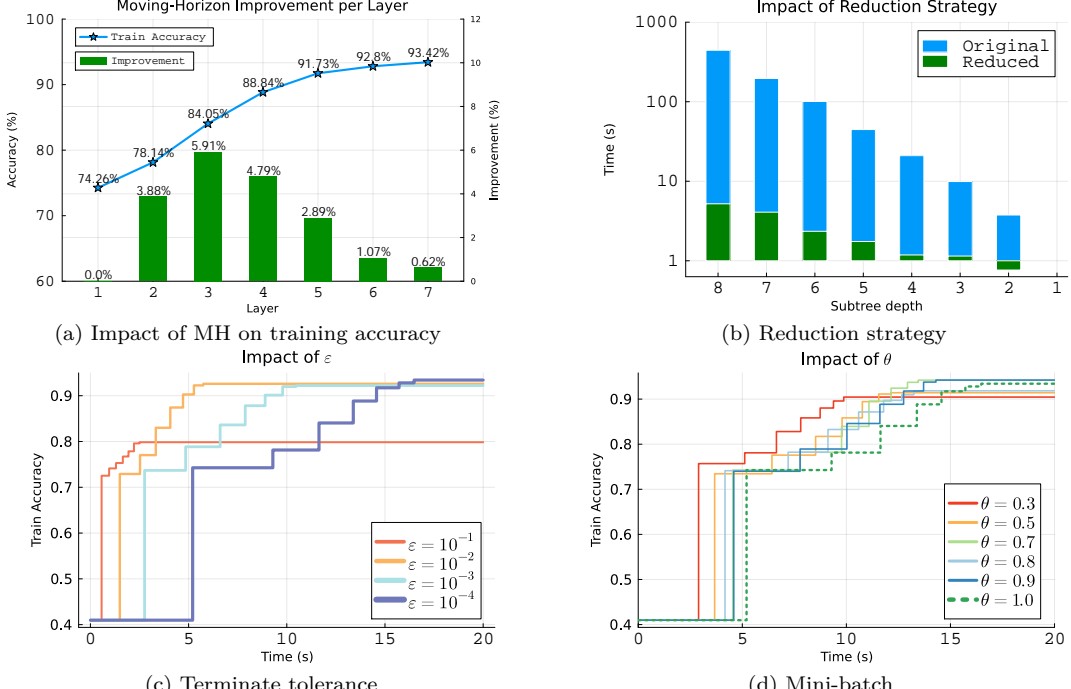

(a) Impact of MH on training accuracy

(b) Reduction strategy

(c) Terminate tolerance

(d) Mini-batch

Figure 8: Impact of the MH procedure, reduction strategy, and parameters $\varepsilon$ and $\theta$. We train a depth-8 tree for `Avila`, where (c) and (d) systematically evaluate MH performance for different subtree depths. For instance, the depth-8 subtree represents the complete original tree, while the depth-7 subtrees are rooted at nodes 2 and 3. Similarly, the depth-6 subtrees are rooted at nodes 4, 5, 6, and 7, and this pattern continues for the other depths.

the most significant improvement occurs within the first five layers. This suggests that the MH refinements have a greater impact on earlier branch nodes, where the corresponding subtrees are relatively larger and contain more samples.

Figure 8(b) demonstrates that the reduction strategy significantly accelerates the computation of subtrees at various depths within the MH procedure. It is evident that the reduction strategy yields a decrease in runtime of nearly two orders of magnitude for the depth-8 subtree. While this reduction in computational time becomes less pronounced as the subtree depth decreases, it still contributes to a considerable overall cost reduction.

Figure 8(c) illustrates the convergence of results obtained through iterative refinement using MH under varying termination conditions. The final accuracies achieved with $\varepsilon = \{10^{-2}, 10^{-3}, 10^{-4}\}$ are notably similar,

but $\varepsilon = 10^{-2}$ results in a 60% reduction in running time. The scenario with $\varepsilon = 10^{-1}$ exhibits a discernible decrease in accuracy. Overall, larger values of $\varepsilon$ lead to lower accuracy for MHABR, but with the benefit of decreased training time. This suggests the possibility of identifying a suitable $\varepsilon$ that incurs a minor reduction in accuracy while yielding significant savings in computational time, as exemplified by $\varepsilon = 10^{-2}$ in this case.

The influence of batch sizes is explored in Figure 8(d). Utilizing mini-batches involves training a model on a subset of the data, which invariably reduces computational time. Interestingly, this approach can sometimes yield superior performance. As depicted in the figure, the accuracies achieved with batch size ratios ($\theta$) of 0.8 and 0.9 surpass those obtained when using the complete dataset. Conversely, a batch size ratio of $\theta = 0.3$ results in a relatively discernible reduction in accuracy (but less than 5%).

### 5.7 Practical Considerations and Limitations

**Class imbalance and multiclass settings.** MHABR applies directly to both binary and multiclass classification because the feature-threshold search is independent of the number of classes, while each leaf predicts the class that minimizes its misclassification loss. Our experiments include multiclass datasets such as Avila, Shuttle, and WESAD, which demonstrate the applicability of the method beyond binary classification. Nevertheless, the unweighted 0-1 objective used in this study may favor majority classes on highly imbalanced datasets, and overall accuracy may not fully reflect minority-class performance. This limitation can be addressed by replacing the loss with a class-weighted objective, $L_w(\mathcal{I}, T) = \sum_{i \in \mathcal{I}} w_{y_i} \mathbf{1}\{y_i \neq T(x_i)\}$, while retaining the same MHABR framework. For problems with many classes, the feature-threshold search space does not directly increase with the number of classes, although deeper trees may be required because a depth-$d$ binary tree has at most $2^d$ leaves and can therefore predict at most $2^d$ distinct class labels.

**Broader Impact** MHABR retains the explicit rule-based structure of decision trees, which can improve transparency and facilitate model auditing. However, interpretability alone does not guarantee fairness, robustness, or safe use. In sensitive applications, the model should be carefully evaluated for bias, distribution shift, and subgroup performance, and should be used with appropriate human oversight.

**Practical Guidance** MHABR is intended for applications in which obtaining a high-quality tree at a prescribed, interpretable depth is important and additional offline training time is acceptable. It is particularly useful when greedy methods such as CART provide insufficient validation accuracy, while global optimization methods are computationally infeasible at the required dataset size or tree depth. We recommend a staged workflow. First, train CART at the prescribed depth to obtain a low-cost accuracy benchmark and a feasible warm start for MHABR. If CART already achieves satisfactory validation performance, further optimization may be unnecessary. Otherwise, apply a light version of MHABR, and compare its validation gain against the additional training time. A more intensive MHABR configuration should be used only when the light version provides a meaningful improvement, as in Avila; when the gain is marginal or absent, as in Htru, CART or DPDT offers a better accuracy-runtime tradeoff.

## 6  Conclusion and Discussion

We introduced MHABR, an approximate method for training deep classification trees within a hierarchical root-subtree framework. The method combines a root-level branch-and-reduce search, an approximate subtree solver based on CART, and a moving-horizon refinement procedure. The resulting algorithm is exact for depth-2 trees and performs well empirically on the benchmark datasets considered in this paper. In particular, it achieves strong test accuracy relative to the compared baselines while remaining computationally feasible on larger datasets than the exact global methods included in our study. These results suggest that MHABR offers a favorable empirical trade-off between optimization quality, runtime, and scalability.

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

# Appendix

# 7 Summary of Notation

Table 7 summarizes the notation used throughout the paper.

Table 7: Summary of notation used in this paper.

| Symbol | Description |
|---|---|
| *Data and decision-tree notation* | |
| $\mathcal{X} = \{x_i\}_{i=1}^n$ | Set of feature vectors, where $x_i \in \mathbb{R}^p$. |
| $\mathcal{Y}$ | Set of class labels. |
| $n$, $p$, $n_c$ | Numbers of samples, features, and classes, respectively. |
| $[n]$ | Index set $\{1, \ldots, n\}$. |
| $\mathcal{I} \subseteq [n]$ | Index set of samples in a tree or subtree. |
| $d$ | Prespecified depth of a decision tree. |
| $\mathcal{T}_d$ | Set of feasible decision trees of depth $d$. |
| $\mathcal{N}_B$, $\mathcal{N}_L$ | Sets of branch nodes and leaf nodes, respectively. |
| $T = [a, k, T_L, T_R]$ | DT with root feature $a$, threshold index $k$, left and right subtrees $T_L$ and $T_R$. |
| **a**, **b**, **c** | Vectors of split features, split thresholds, and leaf predictions. |
| $L(\mathcal{I}, T)$ | Misclassification loss of tree $T$ on the samples indexed by $\mathcal{I}$. |
| $V_d(\mathcal{I})$ | Minimum loss achievable by a depth-$d$ tree on $\mathcal{I}$. |
| $V_d(\mathcal{I}, a, k)$ | Minimum loss when the root feature and threshold index are fixed to $a$ and $k$. |
| $V_0(\mathcal{I})$ | Minimum loss of a constant-label prediction on $\mathcal{I}$. |
| *Candidate splits and sample partitions* | |
| $\mathcal{I}^a$ | Sample indices sorted according to feature $a$. |
| $n_a$ | Number of unique values of feature $a$. |
| $u_k^a$ | The $k$th unique value of feature $a$. |
| $\beta_k^a$ | Candidate split threshold associated with index $k$ for feature $a$. |
| $\mathcal{K}^a$ | Set of candidate threshold indices for feature $a$. |
| $\mathcal{B}^a$ | Set of candidate thresholds $\{\beta_k^a : k \in \mathcal{K}^a\}$. |
| $\zeta(k)$ | The number of samples assigned to the left child by $\beta_k^a$. |
| $\mathcal{I}_{[k_1,k_2]}^a$ | Samples whose values of feature $a$ lie between the thresholds indexed by $k_1$ and $k_2$. |
| *Branch-and-reduce notation* | |
| $\mathcal{K}_i = \{l, \ldots, m\}$ | Candidate threshold-index region represented by a search node. |
| $\bar{k}$ | Midpoint threshold index selected from $\mathcal{K}_i$. |
| $\mathbb{K}$ | Collection of active candidate regions in the branch-and-reduce search. |
| $U$ | Incumbent upper bound, i.e., the best feasible loss found so far. |
| $LB$ | Lower bound associated with a candidate region. |
| $\delta(\bar{k})$ | Reduction radius calculated from the evaluated loss and incumbent upper bound. |
| $\Delta$ | Set of candidate threshold indices removed by the reduction rule. |
| $\mathcal{K}_L$, $\mathcal{K}_R$ | Left and right candidate regions generated after reduction and branching. |
| tol | Optimality-gap tolerance used as a stopping criterion. |
| *Approximation and moving-horizon notation* | |
| $\mathrm{CART}_d(\mathcal{I})$ | Depth-$d$ tree returned by `CART` on the samples indexed by $\mathcal{I}$. |
| $W_d(\mathcal{I})$ | Objective value attained by the `CART` tree. |
| $\widehat{V}_d(\mathcal{I})$ | Optimal value of the `CART`-based approximate child-subtree problem. |
| $\widehat{V}_d(\mathcal{I}, a, k)$ | Approximate objective value when the root split $(a, k)$ is fixed. |
| $t$ | Index of the node used as the root of the current moving-horizon subproblem. |

| | |
|---|---|
| $\mathcal{I}_t$ | Samples reaching node $t$. |
| $\mathcal{I}_{\text{sub}}$ | Sample set associated with the current subtree optimization problem. |
| $T_{\text{sub}}$ | Tree returned for the current subtree optimization problem. |
| $d_{\text{sub}}$ | Depth of the current subtree. |
| $U_{\text{sub}}$ | Objective value of the current subtree solution. |

*Complexity and experimental parameters*

| | |
|---|---|
| $\widetilde{n}$ | Upper bound on the number of candidate splits considered in a subproblem. |
| $\theta \in (0,1]$ | Mini-batch sampling ratio used in the light version of `MHABR`. |
| $\varepsilon$ | Threshold-search termination tolerance used for large-scale datasets. |
| $\alpha$ | Tree-complexity regularization coefficient. |

## 8 Supplementary Experiments: Comparison with TAO

We also provide an informal comparison with the well-known heuristic method `TAO` (Carreira-Perpinán & Tavallali, 2018). Because the authors do not provide publicly available code for reproducibility, we evaluated our method on the same five datasets reported in Appendix 2.1 of their paper, using the identical data split (50% training, 25% validation, and 25% testing). The corresponding results are presented below:

Table 8: Performance comparison between `TAO` and `MHABR`. Results are reported as mean accuracy with standard deviation in parentheses. Best results for each dataset, depth, and split are highlighted in bold.

| Dataset | Depth | Train | | Test | |
|---|---|---|---|---|---|
| | | TAO | MHABR | TAO | MHABR |
| Balance-scale | $d = 2$ | 72.5 (1.6) | **72.7** (0.7) | **69.5** (2.9) | 68.5 (2.0) |
| | $d = 3$ | **76.9** (1.1) | 76.8 (1.4) | 71.6 (1.9) | **74.8** (2.2) |
| | $d = 4$ | 84.0 (1.5) | **84.1** (1.1) | **79.8** (3.1) | 78.9 (2.7) |
| Banknote-auth. | $d = 2$ | 91.9 (0.4) | **92.8** (0.4) | 90.6 (0.9) | **91.3** (1.1) |
| | $d = 3$ | 96.0 (0.5) | **98.3** (0.4) | 95.7 (1.2) | **97.2** (0.6) |
| | $d = 4$ | 98.9 (0.7) | **99.5** (0.3) | 97.2 (0.7) | **98.7** (0.5) |
| Blood-transfusion | $d = 2$ | **78.0** (0.8) | 77.9 (1.4) | 75.8 (2.0) | **77.1** (3.7) |
| | $d = 3$ | 79.5 (1.0) | **80.3** (1.4) | 77.0 (2.0) | **77.8** (2.9) |
| | $d = 4$ | **81.6** (1.3) | 80.7 (2.3) | 77.2 (1.3) | **77.6** (3.5) |
| Breast-cancer | $d = 2$ | 95.0 (0.5) | **96.4** (0.4) | 92.7 (2.2) | **93.9** (1.7) |
| | $d = 3$ | 97.0 (0.6) | **98.5** (0.5) | 93.1 (1.4) | **95.0** (1.7) |
| | $d = 4$ | 98.0 (0.5) | **99.6** (0.3) | 93.2 (0.5) | **95.1** (1.8) |
| Spambase | $d = 2$ | 86.5 (0.7) | **87.3** (0.5) | 86.1 (1.0) | **86.9** (0.6) |
| | $d = 3$ | 90.0 (0.4) | **90.8** (0.3) | 89.1 (1.0) | **90.3** (0.8) |
| | $d = 4$ | 91.8 (0.3) | **92.3** (0.6) | 90.3 (0.8) | **91.3** (0.8) |

Across the matched dataset-depth combinations reported by `TAO`, `MHABR` achieves an average training accuracy of 88.5%, compared with 87.8% for `TAO`, and an average testing accuracy of 86.3%, compared with 85.3%. These differences correspond to average improvements of 0.7 and 1.0 percentage points in training and testing accuracy, respectively. Although performance varies across individual datasets and tree depths, the aggregate results favor `MHABR`. Results for deeper trees are not included because they were not reported for `TAO`.

**Limitations of the `TAO` comparison**  The `TAO` results are taken directly from its original publication rather than obtained through a controlled reimplementation. Because a public implementation is unavailable, we could not standardize data preprocessing, train–test splits, stopping criteria, implementation details, or

computational hardware across the two methods. We therefore present this comparison only as supplementary evidence and do not rely on it to support the primary empirical claims of this paper.

# 9 Additional Theoretical Results and Analysis

In this section, we recall the CART variant used in our theoretical analysis, define its misclassification loss, introduce the reduced CART algorithm, and discuss conditions under which MHABR improves upon this greedy procedure.

## 9.1 Recall the basics of CART

Let $\mathcal{I}_t$ denote the sample index set at node $t \in \mathcal{N}_B \cup \mathcal{N}_L$. The optimal loss of assigning a constant class prediction to this node is

$$V_0(\mathcal{I}_t) := \min_{c \in \mathcal{Y}} L(\mathcal{I}_t, c) = \min_{c \in \mathcal{Y}} \sum_{i \in \mathcal{I}_t} \mathbf{1}\{y_i \neq c\}. \tag{38}$$

The misclassification-based CART rule selects the feature and threshold pair that minimizes the combined potential loss of its two children:

$$(a'_t, k'_t) \in \arg \min_{a \in [p], \ k \in \mathcal{K}^a} \left\{ V_0\left(\mathcal{I}^a_{t,[0,k]}\right) + V_0\left(\mathcal{I}^a_{t,[k,n_a]}\right) \right\}, \tag{39}$$

where $\mathcal{I}^a_{t,[0,k]}$ and $\mathcal{I}^a_{t,[k,n_a]}$ are the sample sets assigned to the left and right children, respectively.

For a sample index set $\mathcal{I}$, let $\hat{T}_d(\mathcal{I})$ denote the depth-$d$ tree returned by this recursive CART procedure. Its loss is defined as the resulting misclassification count:

$$W_d(\mathcal{I}) := L\left(\mathcal{I}, \hat{T}_d(\mathcal{I})\right) = \sum_{i \in \mathcal{I}} \mathbf{1}\left\{y_i \neq \hat{T}_d(x_i)\right\}. \tag{40}$$

For $d = 0$, the tree consists of a majority-class leaf, and therefore

$$W_0(\mathcal{I}) = V_0(\mathcal{I}). \tag{41}$$

If $(a', k')$ denotes the root split selected by CART, then the additivity of the misclassification loss gives

$$W_d(\mathcal{I}) = W_{d-1}\left(\mathcal{I}^{a'}_{[0,k']}\right) + W_{d-1}\left(\mathcal{I}^{a'}_{[k',n_{a'}]}\right). \tag{42}$$

The quantity $W_d(\mathcal{I})$ always denotes the final misclassification loss of the resulting tree. In the numerical comparison, the standard CART baseline may use a different impurity criterion, such as entropy, to select its splits; this does not change the definition of its reported misclassification loss.

**Depth-1 optimality** For a sample subset $\mathcal{I} \subseteq [n]$, a depth-1 DT comprises two optimal constant fits, i.e., two optimal depth-0 subtrees, expressed by $V_0(\mathcal{I}) = \min_{c \in \mathcal{Y}} L(\mathcal{I}, c)$, which denotes the loss of the best constant approximation to $\mathcal{Y}$, in the same manner as CART. For $d = 1$, CART solves a one-stage decision problem, which, according to the greedy nature, is optimal.

**Monotonicity** The loss function of CART exhibits monotonicity with respect to tree depth, such that $W_d(\mathcal{I}) \leq W_{d-1}(\mathcal{I})$. This is attributed to the monotonic nature of the branching operation concerning the count of correctly classified samples (accuracy). Branching will either increase this count (thereby decreasing the loss) or, in the worst-case scenario where the newly formed leaves offer no improvement over their parent node, the count will remain unchanged.

For example, we consider a node $t$ with data $\mathcal{I}_t$, the potential loss is $V_0(\mathcal{I}_t)$ as defined by Equation (38), with prediction $c_1$. Then, we branch the node to obtain two branch nodes $\mathcal{I}_{2t}, \mathcal{I}_{2t+1}$, where $\mathcal{I}_{2t} \cup \mathcal{I}_{2t+1} = \mathcal{I}_t$. The

losses of these new nodes are $V_0(\mathcal{I}_{2t})$ and $V_0(\mathcal{I}_{2t+1})$ with predictions $c_2$ and $c_3$, respectively. Then we have:

$$
\begin{aligned}
V_0(\mathcal{I}_t) = \sum_{i \in \mathcal{I}} \mathbb{1}\{y_i \neq c_1\} &= \sum_{i \in \mathcal{I}_{2t}} \mathbb{1}\{y_i \neq c_1\} + \sum_{i \in \mathcal{I}_{2t+1}} \mathbb{1}\{y_i \neq c_1\} \\
&\geq \sum_{i \in \mathcal{I}_{2t}} \mathbb{1}\{y_i \neq c_2\} + \sum_{i \in \mathcal{I}_{2t+1}} \mathbb{1}\{y_i \neq c_3\} \\
&= V_0(\mathcal{I}_{2t}) + V_0(\mathcal{I}_{2t+1}),
\end{aligned}
\tag{43}
$$

The inequality holds because $c_2$ and $c_3$ are chosen as the predictions that minimize the misclassifications within their respective nodes $2t$ and $2t+1$ (defined as the most frequent class in Equation (38)). By definition, the optimal prediction $c_2$ for node $\mathcal{I}_{2t}$ must classify at least as many samples correctly within $\mathcal{I}_{2t}$ as any other prediction, including $c_1$. Thus, $\sum_{i \in \mathcal{I}_{2t}} \mathbb{1}\{y_i \neq c_2\} \leq \sum_{i \in \mathcal{I}_{2t}} \mathbb{1}\{y_i \neq c_1\}$, and similarly for $c_3$ over $\mathcal{I}_{2t+1}$. Therefore, the loss after branching is less than or equal to the value before branching.

### 9.2 A reduced `CART` method

---
**Algorithm 4** Reduced `CART` for a depth-$d$ tree

---
1: **function** R-CART$(\mathcal{I}, d)$
2:     Compute $V_0(\mathcal{I}) \leftarrow \min_{c \in \mathcal{Y}} L(\mathcal{I}, c)$ and $c' \leftarrow \arg\min_{c \in \mathcal{Y}} L(\mathcal{I}, c)$
3:     **if** $d = 0$ **or** $|\mathcal{I}| \leq 1$ **or** $V_0(\mathcal{I}) = 0$ **then**
4:       **return:** $V_0(\mathcal{I})$ and leaf $c'$
5:     Initialize $U \leftarrow +\infty$ and $(a', k') \leftarrow \emptyset$
6:     **for** $a \in [p]$ **do**
7:       Sort $\mathcal{I}$ by feature $a$ to obtain $\mathcal{I}^a$ and $\mathcal{K}^a$, set $\mathcal{K} \leftarrow \mathcal{K}^a$
8:       **while** $\mathcal{K} \neq \emptyset$ **do**
9:         Select $\bar{k} \leftarrow \mathcal{K}[1]$ and compute $V_1(\mathcal{I}, a, \bar{k}) \leftarrow V_0(\mathcal{I}^a_{[0,\bar{k}]}) + V_0(\mathcal{I}^a_{[\bar{k},n_a]})$
10:         **if** $V_1(\mathcal{I}, a, \bar{k}) < U$ **then**
11:           Update $U \leftarrow V_1(\mathcal{I}, a, \bar{k})$ and $(a', k') \leftarrow (a, \bar{k})$
12:         Compute $\delta(\bar{k}) \leftarrow V_1(\mathcal{I}, a, \bar{k}) - U$ and get $\mathcal{K} \leftarrow \{k \in \mathcal{K} \mid \zeta(k) - \zeta(\bar{k}) > \delta(\bar{k})\}$
13:     Get the data $\mathcal{I}_L \leftarrow \mathcal{I}^{a'}_{[0,k']}$ and $\mathcal{I}_R \leftarrow I^{a'}_{[k',n_{a'}]}$
14:     Compute $(W_{d-1}(\mathcal{I}_L), T_L) \leftarrow$ R-CART$(\mathcal{I}_L, d-1)$ and $(W_{d-1}(\mathcal{I}_R), T_R) \leftarrow$ R-CART$(\mathcal{I}_R, d-1)$
15:     Calculate $W_d(\mathcal{I}) \leftarrow W_{d-1}(\mathcal{I}_L) + W_{d-1}(\mathcal{I}_R)$ and update $T \leftarrow [a', k', T_L, T_R]$
16:     **return:** Loss $W_d(\mathcal{I})$ and decision tree $T$

---

At each nonterminal node while using `BR` method, `CART` solves a depth-1 split-selection problem by evaluating $V_1(\mathcal{I}, a, k)$ over the candidate feature-threshold pairs. When $d = 1$, the child-subtree problems reduce to optimal constant predictions, and hence `ApproxTreeSearch`$(\mathcal{I}, 1)$ is exact. In this setting, the reduction strategy of Algorithm 2 can be applied directly to the local `CART` split search. We refer to this reduced implementation as `R-CART`, whose procedure is given in Algorithm 4.

Unlike the midpoint-based branching strategy used in `BR`, `R-CART` scans the candidate thresholds in ascending order and skips candidates that are certified not to improve the current incumbent. The reduction is applied only to the local depth-1 split objective. After a split is selected, `R-CART` is recursively applied to the two child datasets to construct a tree of depth $d$. Under the same candidate sets, stopping conditions, and tie-breaking rule, `R-CART` returns the same tree as standard `CART`, while potentially evaluating fewer split candidates.

Because `MHABR` repeatedly calls `CART` when approximating child-subtree problems, the computational savings provided by `R-CART` can accumulate over the entire algorithm. Without reduction, the root-level search evaluates at most $\sum_{a=1}^{p} |\mathcal{K}^a|$ candidate splits. The reduction strategy decreases this number by skipping candidate thresholds that cannot strictly improve the incumbent solution.

### 9.3 An Illustrative XOR Case for Greedy Failure

In this section, we use XOR-style datasets as an illustrative example to explain why a lookahead-based method can outperform greedy `CART`. The XOR problem is a canonical topological trap for greedy splitting: a depth-1 criterion may fail to identify informative features, whereas a shallow lookahead can recover the correct interaction. Our goal here is to build intuition for the behavior of `MHABR`.

More broadly, `ABR` can improve upon `CART` in settings where a split that appears suboptimal under a purely greedy criterion leads to a better downstream subtree configuration under lookahead. The following XOR example provides a simple instance of this phenomenon.

**XOR-Distractor Dataset.** We consider the following stylized data distribution.

**Definition 9.1.** Let the dataset be given by $\mathcal{X} = \{(x_i, y_i)\}_{i=1}^n$ where $x_i \in \{0,1\}^p$ and $y_i \in \mathcal{Y} = \{0,1\}$. The target $y_i$ is determined exclusively by the first two features via the XOR function: $y_i = x_i^1 \oplus x_i^2$, where $x_i^1$ and $x_i^2$ are drawn from independent uniform Bernoulli distributions ($P = 0.5$). The remaining features $x_i^3, \ldots, x_i^p$ are pure noise, drawn independently from $P(x_i^h = 1) = 0.5$ for $h \geq 3$.

In this setting, a single greedy split on either informative feature does not reduce the immediate misclassification loss, because each informative feature alone leaves the label distribution balanced within the resulting child nodes. In contrast, a depth-2 lookahead can recover the XOR interaction by selecting splits that jointly isolate the homogeneous quadrants.

**Theorem 9.2** (Failure of Greedy Splits). *Given an XOR-Distractor dataset defined in Definition 9.1, let $V_0(\mathcal{I})$ denote the misclassification loss (obtained using a constant prediction) of a root node containing $\mathcal{I}$ with $n$ samples. Suppose that, in the realized finite sample, the classes are balanced at the root and remain balanced in both child nodes after splitting on either true feature $a = 1$ or $a = 2$. Suppose further that there exists a noise feature $h \geq 3$ for which the class distribution is not balanced in at least one of the resulting child nodes. Then, a greedy top-down decision tree algorithm that selects a root split by maximizing the immediate misclassification-loss reduction will select a noise feature rather than either true feature $a = 1$ or $a = 2$.*

*Proof.* Let $|\mathcal{I}| = n$. Since the classes are balanced at the root, $V_0(\mathcal{I}) = 0.5n$. For an informative feature $a \in \{1, 2\}$, let $\mathcal{I}_L^a$ and $\mathcal{I}_R^a$ denote the two child-node sample sets. By assumption, both child nodes remain class-balanced. Hence, $V_0(\mathcal{I}_L^a) = 0.5|\mathcal{I}_L^a|$ and $V_0(\mathcal{I}_R^a) = 0.5|\mathcal{I}_R^a|$. Therefore,

$$\Delta V_0(a = 1) = V_0(\mathcal{I}) - \big(V_0(\mathcal{I}_L^1) + V_0(\mathcal{I}_R^1)\big)$$
$$= 0.5n - \big(0.5|\mathcal{I}_L^1| + 0.5|\mathcal{I}_R^1|\big)$$
$$= 0.5n - 0.5n = 0$$

Now consider the noise feature $h \geq 3$. For $s \in \{L, R\}$ and $c \in \{0, 1\}$, define $n_{s,c}^h := |\{i \in \mathcal{I}_s^h : y_i = c\}|$. The minimum constant-prediction loss in child node $\mathcal{I}_s^h$ is $V_0(\mathcal{I}_s^h) = \min\{n_{s,0}^h, n_{s,1}^h\} = (|\mathcal{I}_s^h| - |n_{s,1}^h - n_{s,0}^h|)/2$. Thus,

$$\Delta V_0(a = h) = (|n_{L,1}^h - n_{L,0}^h| + |n_{R,1}^h - n_{R,0}^h|)/2$$

By assumption, at least one child node is class-unbalanced, so $\Delta V_0(h) > 0$. Therefore,

$$\Delta V_0(a = h) > \Delta V_0(a = 1) = \Delta V_0(a = 2) = 0,$$

and a greedy algorithm maximizing immediate loss reduction selects a noise feature rather than either informative feature. $\square$

**Illustrative role of lookahead** For XOR-style data, a depth-2 tree can represent the target rule exactly, whereas a purely greedy depth-1 criterion may not discover the required structure. This makes XOR an example for visualizing the difference between greedy splitting and lookahead. Figure 9 illustrates this contrast.

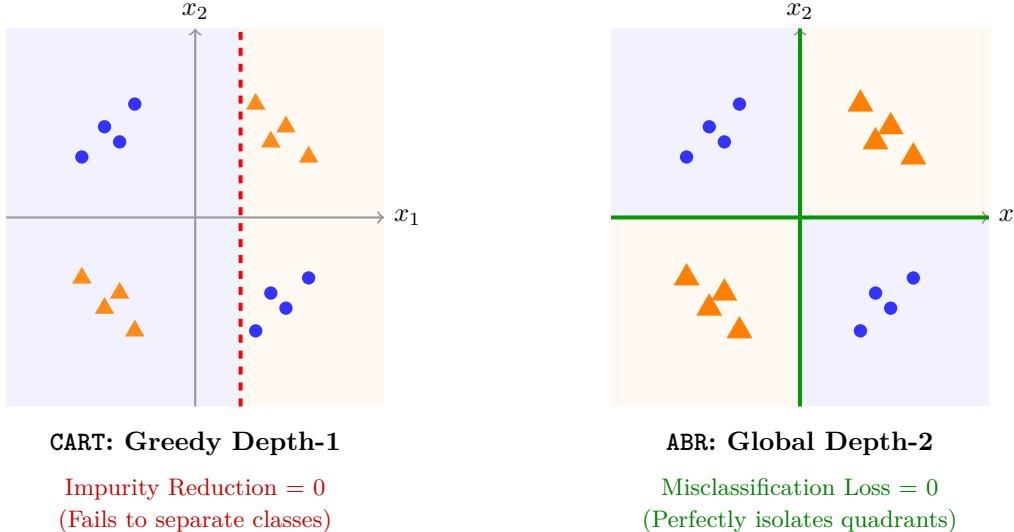

Figure 9: Illustration of a canonical XOR failure mode. A depth-1 greedy split may fail to separate the classes, whereas a depth-2 lookahead can recover the correct interaction structure.

This example highlights a basic limitation of greedy splitting: the usefulness of a feature may only become apparent after subsequent splits are taken into account. A lookahead-based method such as `ABR` is designed to evaluate such downstream effects more explicitly.

**Theorem 9.3.** *Let $\mathcal{D}$ be the XOR-distributed dataset described above, where the two informative features are binary. For $d \geq 2$, `MHABR` returns a global optimal tree with zero training loss, even when the reduction strategy is enabled. Moreover, if `CART` returns a tree with positive loss, as established in Theorem 9.2, then `MHABR` strictly outperforms `CART`.*

*Proof.* Let $W_d(\mathcal{I})$ denote the misclassification loss of the initial tree generated by `CART`, establishing the initial upper bound $U = W_d(\mathcal{I})$ for `MHABR`. Let $a \in \{1, 2\}$ be an informative feature and let $k^*$ denote its nontrivial split index. We aim to prove that $k^*$ is never removed by the reduction strategy defined in Lemma 3.3 before a globally optimal solution is found.

By the definition of XOR, splitting on feature $a$ at $k^*$ and then on the other informative feature partitions the sample space into four class-pure quadrants. Therefore, the corresponding depth-2 tree has zero misclassification loss: $V_2(\mathcal{I}, a, k^*) = 0$. For every $d \geq 2$, the training loss of `CART` is nonincreasing with the available depth, and a depth-1 tree is solved exactly. Hence, for every candidate split $(a, k)$,

$$\widehat{V}_d(\mathcal{I}, a, k) = W_{d-1}(\mathcal{I}^a_{[0,k]}) + W_{d-1}(\mathcal{I}^a_{[k,n_a]})$$
$$\leq W_1(\mathcal{I}^a_{[0,k]}) + W_1(\mathcal{I}^a_{[k,n_a]})$$
$$= V_2(\mathcal{I}, a, k).$$

Since the loss is nonnegative and $V_2(\mathcal{I}, a, k^*) = 0$, it follows that $\widehat{V}_d(\mathcal{I}, a, k^*) = 0$. During the search, suppose that the algorithm evaluates a midpoint $\bar{k}$ before evaluating $k^*$. Applying Lemma 3.2 to the exact depth-2 objective gives $V_2(\mathcal{I}, a, \bar{k}) = |V_2(\mathcal{I}, a, \bar{k}) - V_2(\mathcal{I}, a, k^*)| \leq |\zeta(\bar{k}) - \zeta(k^*)|$. Consequently,

$$\widehat{V}_d(\mathcal{I}, a, \bar{k}) \leq V_2(\mathcal{I}, a, \bar{k}) \leq |\zeta(\bar{k}) - \zeta(k^*)|. \tag{44}$$

Before a zero-loss solution is found, the incumbent satisfies $U > 0$. According to the reduction strategy, the reduction margin is $\delta(\bar{k}) := \widehat{V}_d(\mathcal{I}, a, \bar{k}) - U$. Therefore,

$$\delta(\bar{k}) = \widehat{V}_d(\mathcal{I}, a, \bar{k}) - U < \widehat{V}_d(\mathcal{I}, a, \bar{k}).$$

Combining this inequality with Equation (44) yields

$$\delta(\bar{k}) < |\zeta(\bar{k}) - \zeta(k^*)|.$$

Hence, the condition required to remove $k^*$, $|\zeta(\bar{k}) - \zeta(k^*)| \leq \delta(\bar{k})$ cannot hold. Thus, $k^*$ is not removed by the reduction strategy. If $U = 0$ at an earlier iteration, a global optimal solution has already been found. Since the candidate set is finite and $k^*$ remains active, it is eventually evaluated, after which the incumbent is updated to $U_{\texttt{MHABR}}(\mathcal{I}, d) = V_d(\mathcal{I}) = 0$. Because misclassification loss is nonnegative, this solution is global optimal. Finally, if the initial `CART` tree satisfies $W_d(\mathcal{I}) > 0$, then $U_{\texttt{MHABR}}(\mathcal{I}, d) = 0 < W_d(\mathcal{I})$, so `MHABR` strictly outperforms `CART`. $\square$

## 10 Details of Numerical Experiments

### 10.1 Information of small-scale datasets

The details of the 51 small datasets are presented in Table 9.

Table 9: The information of 51 small-scale datasets.

| Dataset Name | $n$ | $p$ | Class | Dataset Name | $n$ | $p$ | Class |
|---|---|---|---|---|---|---|---|
| Soybean-small | 47 | 35 | 4 | Body | 507 | 5 | 2 |
| Echocardiogram | 61 | 11 | 2 | Climate-model-crashes | 540 | 20 | 2 |
| Hepatitis | 80 | 19 | 2 | Monks-problems-3 | 554 | 6 | 2 |
| Fertility | 100 | 9 | 2 | Monks-problems-1 | 556 | 6 | 2 |
| Acute-inflammations-1 | 120 | 6 | 2 | Breast-cancer-diagnosti | 569 | 30 | 2 |
| Acute-inflammations-2 | 120 | 6 | 2 | Monks-problems-2 | 600 | 6 | 2 |
| Hayes–roth | 132 | 5 | 3 | Balance-scale | 625 | 4 | 3 |
| Iris | 150 | 4 | 3 | Credit-approval | 653 | 15 | 2 |
| Teaching-assistant-evaluation | 151 | 5 | 3 | Breast-cancer | 683 | 9 | 2 |
| Wine | 178 | 13 | 3 | Blood-transfusion | 748 | 4 | 2 |
| Breast-cancer-prognostic | 194 | 31 | 2 | Mammographic-mass | 830 | 5 | 2 |
| Parkinsons | 195 | 23 | 2 | Tic-tac-toe-endgame | 958 | 9 | 2 |
| Connectionist-bench-sonar | 208 | 60 | 2 | Connectionist-bench | 990 | 13 | 11 |
| Image-segmentation | 210 | 19 | 7 | Statlog-project-German-credit | 1,000 | 20 | 2 |
| Seeds | 210 | 7 | 3 | Concrete | 1,030 | 8 | 3 |
| Glass | 214 | 9 | 6 | Banknote-authentication | 1,372 | 4 | 2 |
| Thyroid-disease-new-thyroid | 215 | 5 | 3 | Contraceptive-method-choice | 1,473 | 9 | 3 |
| Congressional-voting-records | 232 | 16 | 2 | Car-evaluation | 1,728 | 6 | 4 |
| Spect-heart | 267 | 22 | 2 | Ozone-level-detection-eight | 1,847 | 72 | 2 |
| Spectf-heart | 267 | 44 | 2 | Ozone-level-detection-one | 1,848 | 72 | 2 |
| Cylinder-bands | 277 | 39 | 2 | Seismic-bumps | 2,584 | 18 | 2 |
| Heart-disease-Cleveland | 282 | 13 | 5 | Chess-king-rook-versus-king-pawn | 3,196 | 36 | 2 |
| Haberman-survival | 306 | 3 | 2 | Thyroidann | 3,772 | 21 | 3 |
| Ionosphere | 351 | 34 | 2 | Wall-following-robot-2 | 5,456 | 2 | 4 |
| Dermatology | 358 | 34 | 6 | Thyroid-disease-ann-thyroid | 7,200 | 21 | 3 |
| Thoracic-surgery | 470 | 16 | 2 | | | | |

### 10.2 Configuration of additional techniques for large-scale datasets

The configurations for datasets containing more than $10,000$ are described below. For the 5 medium-scale datasets, we do not introduce the parameters $\varepsilon$ and $\theta$; `MHABR` successfully completes all the training tasks. For the three large-scale datasets, we configure the algorithms to terminate within the time limit. The configurations are SUSY: $\varepsilon = 0.01, \theta = 0.25$, HIGGS: $\varepsilon = 0.01, \theta = 0.5$, WESAD: $\varepsilon = 0.00025, \theta = 0.25$.

### 10.3 Implementation details

Details and experimental settings of all comparison algorithms are stated below. Unless otherwise specified, the implementations used in our experiments are obtained from their original authors.

**MHABR**: Our algorithm is implemented in `Julia`. For the default version, we adopt the reduction strategy with parameters $\varepsilon = 0$ and $\theta = 1$.

**CART** (Sadeghi et al., 2022): We use the implementation from the `Julia` package `DecisionTree`, selecting entropy loss as it provides the best results among the three options; its performance may occasionally surpass `MHABR` and `DPDT` on several datasets.

**DPDT** (Kohler et al., 2025): This algorithm is written in `Python` and calls `CART` through the `Python` package `scikit-learn`, using the **Gini loss** (default). Therefore, in our results, it may be surpassed by `CART`.

**LS-OCT** (Dunn, 2018): Since the original code is not available, we implement both methods in `Julia` and call `Gurobi` to solve MIP models.

**DL8.5** (Aglin et al., 2020): This algorithm is written in `C++` and is run as an extension of `Python`. The current version also integrates the methods of `MurTree` (Demirović et al., 2022). Besides, we use `GUESS` (McTavish et al., 2022) for binarization because it provided the best performance among the evaluated options.

**Quant-BnB** (Mazumder et al., 2022): The authors provide an open-source implementation of this algorithm, which is written in `Julia`.

**TAO** (Carreira-Perpinán & Tavallali, 2018): Because the source code for this method is not publicly available, we directly compare our approach with the results reported in the original paper under the same experimental configuration.

Additionally, we experimented with **STreeD** (van der Linden et al., 2023), implemented in `C++` with a `Python` interface. However, due to changes in computational hardware, a fair comparison could not be ensured; therefore, we omit its results.

