# OpenReview forum: "A Moving-Horizon Approximate Branch-and-Reduce Method for Deep Classification Trees"
_TMLR — Under review for TMLR_

### Review · Reviewer_wLdC · 2026-05-14

**Summary Of Contributions:**

This paper proposes an efficient method for computing high-accuracy decision trees. The approach first decomposes the training process into an upper-level and a lower-level problem, where the upper level is solved exactly, while a CART-based approximation is introduced for the lower level problem. Furthermore, the method is made efficient by combining this approximate model with a branch-and-bound-based pruning strategy (ABR). In the proposed approach, the decision tree is iteratively refined by repeatedly applying ABR while traversing the nodes. In the experiments, benchmark datasets are categorized by size and evaluated, demonstrating both high accuracy and computational efficiency.

Strength:

- The experimental performance appears to be sufficiently strong in terms of both accuracy and efficiency.

- The paper is generally well organized.

Weakness:

- The paper first presents upper and lower bounds, along with pruning based on them, under the assumption of exact computation. However, the final proposed method introduces a CART-based approximation, and it is unclear how the theory of these bounds ultimately contributes to the validity of the proposed method.

- Some parts of the technical description lack sufficient detail, and further clarification would be desirable.

**Audience:**

Yes

**Audience Explanation:**

The decision tree is still a quite important technique in machine learning and thus can attract the audience of TMLR.

**Broader Impact Concerns:**

None.

**Claims And Evidence:**

Yes

**Claims Explanation:**

The experimental results sufficiently support the authors claim.

**Requested Changes:**

I have a few technical questions, but overall, I do not have a severe concern.

- As mentioned in the weaknesses, could you clarify the relationship between the (exact) bound-based theory and the proposed approximation method? In particular, it would be important to understand whether any form of validity, linked to these bounds, still holds in the approximate search after introducing the CART. For example, Lemma 4.1 assumes no reduction. What aspect of the proposed method can be justified by a proof under the no-reduction setting? Is there anything that can be guaranteed in the presence of reduction?

- Is the term “bilevel optimization” appropriate in this context? Typically, bilevel optimization refers to problems of the form max_x f(x,y*(x)) where y*(x) = argmax_y g(x,y). In such formulations, the optimality of a different lower-level problem constrains the upper-level problem. However, the optimization problem in the proposed method does not appear to follow this structure. I found this somewhat confusing and feel that the terminology may not be ideal.

- In Theorem 5.2, it is stated that there are at most \tilde{n} splits. Does this ultimately correspond to O(n)? In other words, can the reduction be explicitly characterized in terms of computational complexity with respect to n?

- Related to the above question, at the very least, since all data points must be examined at least once, it seems unusual that an O(n) term does not appear in the computational complexity of MHABR.

- At the start of Algorithm 4, an initial tree must be prepared; however, I could not find a description of how this initial tree is constructed.

- Since \alpha-tuning is used in Table 3, training accuracy is difficult to interpret. In my understanding, since the algorithm is designed to optimize a penalized objective function, comparing the unpenalized term does not allow us to draw conclusions about the relative performance of the optimization algorithms.

- In Algorithm 2, \delta(\bar{k}) is computed but does not appear to be used in Algorithm. Since \delta seems to be an important component, it would be helpful to explicitly clarify how it is used in the algorithm.

---

> ### Author Response · Authors · 2026-07-03
>
> We thank the reviewer for the positive assessment and constructive technical questions. We have revised the manuscript substantially to clarify the theoretical status, terminology, complexity, initialization, and experimental interpretation.
>
> ### 1. Exact bounds versus the CART-based approximation
> We now distinguish explicitly between the exact BR method and the approximate ABR/MHABR method. Lemmas 3.2--3.3 provide a safe reduction rule for the exact value function $V_d$, where both child-subtree problems are solved globally. Accordingly, Algorithm 1 retains the global-optimality guarantee.
> For Algorithm 2, Lemma 4.1 and Remark 4.2 now state precisely what remains valid after replacing $V_{d-1}$ by the CART loss $W_{d-1}$:
> - with or without reduction, the returned loss is no worse than the CART initialization;
> - without reduction, ABR exhaustively searches all root splits and exactly minimizes the approximate objective $\widehat V_d$;
> - for $d=2$, CART solves each depth-1 child-subtree problem exactly, so $\widehat V_2=V_2$ and the reduction remains safe, yielding a globally optimal depth-2 tree;
> - for $d>2$, we do not claim that reduction preserves the minimizer of $\widehat V_d$ or global optimality; it is explicitly presented as a heuristic acceleration.
> We also added Corollary 4.3, proving that every accepted moving-horizon update decreases the training loss and that the final MHABR loss cannot exceed the CART initialization.
>
> ### 2. Terminology
> We agree that “bilevel optimization” was potentially misleading. We replaced it throughout the manuscript with **hierarchical root-subtree optimization**, and introduced the terms **root-level problem (RLP)** and **child-subtree problem (CSP)**.
>
> ### 3. Meaning of $\widetilde n$ and dependence on $n$
> The revised Theorem 4.4 (formerly Theorem 4.2) defines $\widetilde n$ as the maximum actual number of feature-threshold pairs evaluated in a BR or ABR root search. Since each feature has at most $n+1$ candidate thresholds,
> $$
> \widetilde n\le p(n+1)=O(pn).
> $$
> We now provide both the parameterized bounds and the coarse worst-case bounds obtained by substituting $\widetilde n=O(pn)$.
>
> ### 4. Missing data-processing cost
> We agree that the previous complexity statement omitted the unavoidable cost of processing the samples. The revised theorem explicitly assumes sorting at each processed tree/subtree problem, costing $O(pn_t\log n_t)$. It proves
> $$
> C_{\mathrm{CART}}=O(pnd\log n),\qquad
> C_{\mathrm{BR}}=O(pn\log n\,\widetilde n^{d-1}),
> $$
> and
> $$
> C_{\mathrm{MHABR}}=O\left(\widetilde n\,pn\log n\,\frac{d(d-1)}{2}\right).
> $$
> Thus, the dependence on $n$ is now explicit.
>
> ### 5. Initial tree
> Algorithm 3 (formerly Algorithm 4) now states explicitly that MHABR initializes both the tree $T$ and its loss $U$ using a depth-$d$ CART tree. Algorithm 2 likewise initializes each ABR search with CART.
>
> ### 6. Interpretation of Table 3
> We agree that, after validation-based $\alpha$ selection, unpenalized training accuracy should not be interpreted as a comparison of optimization quality under a common objective. We revised Section 5.3 to state that Table 3 evaluates model selection and predictive performance under a common validation protocol. Testing accuracy is the primary quantity of interest, while training accuracy is reported only descriptively.
>
> ### 7. Use of $\delta(\bar k)$
> Algorithm 2 now shows its use explicitly. After computing$\delta(\bar k)=\widehat V_d(\mathcal I,a,\bar k)-U$,
> the algorithm constructs $\Delta_i=\{k\in K_i:\lvert\zeta(k)-\zeta(\bar k)\rvert\le\delta(\bar k)\},$ removes $\Delta_i$, and branches only on the remaining left and right candidate sets.
>
> We appreciate the reviewer’s comments, which helped us clarify both the guarantees and limitations of the proposed approximation.

---

### Review · Reviewer_CTdj · 2026-05-26

**Summary Of Contributions:**

This paper proposes MHABR (Moving-Horizon Approximate Branch-and-Reduce), a method for training near-optimal deep classification trees on large-scale datasets with continuous features. The core idea is a bilevel optimization framework where the upper-level problem (root node selection) is solved via a branch-and-reduce method, while the lower-level problem (subtree optimization) is approximated using CART. A moving-horizon refinement procedure iteratively re-optimizes subtrees to progressively close the gap to the global optimum. The key contributions are:

A bilevel formulation of decision tree training with a principled branch-and-reduce solver for the upper-level problem
A Lipschitz-type sensitivity lemma enabling an efficient reduction strategy that prunes the candidate split space without discarding optimal solutions
Global optimality guarantees for depth-2 trees, and near-optimal performance at greater depths
Extensive experiments on 59 UCI datasets spanning small, medium, and large scales, demonstrating that MHABR consistently outperforms heuristic baselines (CART, DPDT) and scales beyond what global solvers (Quant-BnB, DL8.5) can handle

Strengths:

Clear theoretical grounding with rigorous proofs
Strong and broad empirical evaluation across dataset scales and tree depths
Practical efficiency gains enabled by the reduction strategy and warm-starting
RL perspective on the lookahead approximation is insightful and well-motivated

Weaknesses:

Comparison with TAO is informal due to the lack of public code, limiting the reproducibility of that portion
Computational overhead of MHABR on some medium datasets (e.g., Htru) is notably high relative to accuracy gains
The paper is a long submission, and some appendix material could be better integrated into the main text

**Additional Comments:**

This is a solid and well-executed paper. The combination of a clean theoretical framework, a practically efficient algorithm, and thorough empirical validation makes it a strong contribution to the literature on optimal and near-optimal decision trees. The moving-horizon strategy is a natural and well-motivated extension of the approximate branch-and-reduce idea, and the reduction strategy is both elegant and impactful in practice. The paper is recommended for acceptance.

**Audience:**

Yes

**Audience Explanation:**

This work is likely to be of interest to multiple segments of the TMLR audience. Researchers working on interpretable machine learning and decision tree induction will find the bilevel optimization framework and the near-optimality guarantees directly relevant. Those working on combinatorial optimization and branch-and-bound methods will appreciate the Lipschitz-type reduction strategy and its generalization to arbitrary depths. Practitioners dealing with large-scale tabular datasets will be interested in the scalability results, particularly the ability to train depth-8 trees on datasets with tens of millions of samples. The reinforcement learning perspective on the lookahead approximation also broadens the appeal to the RL community. Overall, the paper addresses a well-known scalability bottleneck in a practically important model class, and does so with both theoretical and empirical rigor.

**Broader Impact Concerns:**

This work focuses on decision tree induction for classification, which is inherently an interpretability-oriented approach. The improved scalability and accuracy of MHABR could facilitate broader adoption of interpretable models in high-stakes domains such as healthcare, finance, and public policy, where transparency is critical. No significant negative ethical implications are apparent. A brief broader impact statement acknowledging the interpretability benefits and the potential misuse of any classification model in sensitive settings would be a helpful addition if not already present.

**Claims And Evidence:**

Yes

**Claims Explanation:**

The claims are well supported by a combination of theoretical results and empirical evidence. On the theoretical side, the paper provides rigorous proofs of the key sensitivity lemma (Lemma 3.2), the correctness of the reduction strategy (Lemma 3.3), global convergence of the BR method (Theorem 3.4), and the guarantee that MHABR is no worse than CART (Lemma 4.1). The complexity analysis (Theorem 4.2) is clearly derived. On the empirical side, experiments are conducted on 59 datasets across small, medium, and large scales, with multiple baselines and multiple tree depths. Results are averaged over 10 runs. The ablation study in Section 5.6 effectively isolates the contributions of the MH refinement, reduction strategy, and tunable parameters. The optimality gap evaluation in Section 5.5 provides quantitative grounding for the near-optimality claims. Minor limitations include the informal TAO comparison and the absence of statistical significance testing, but these do not undermine the overall convincingness of the evidence.

**Requested Changes:**

The following changes would strengthen the paper, though none are critical blockers given the overall quality of the submission.
Important (would strengthen the paper):

The comparison with TAO relies on self-reported numbers from the original paper, which limits reproducibility. If a public implementation becomes available, a direct comparison would substantially strengthen this section. At minimum, a clearer disclaimer about the limitations of this comparison should be included.
On some medium-scale datasets (e.g., Htru), the runtime of MHABR is substantially higher than DPDT with only modest accuracy gains. A more explicit discussion of when the additional computational cost is and is not justified would help practitioners calibrate expectations.
The paper is classified as a long submission, and some appendix content (particularly Section 8.3 on the XOR case) is closely tied to the main algorithmic intuition. Consider moving key intuition from this section into the main text, or at least adding a forward reference with a brief summary.
A brief discussion of how MHABR behaves under class imbalance, or on datasets with many classes, would help clarify the generality of the approach.

Minor:

Some notation is introduced locally and could benefit from a unified notation table in the appendix.
The interpretation of Figure 3 (log-scale runtime comparison) would benefit from a brief prose summary of the key takeaways in the caption or surrounding text.

---

> ### Author Response · Authors · 2026-07-03
>
> We sincerely thank the reviewer for the positive and constructive assessment. We have revised the manuscript to address all requested changes. The main revisions are summarized below.
>
> **1. Reproducibility of the TAO comparison.**
> We agree that the TAO comparison cannot be regarded as a controlled benchmark because no public implementation is available. We now state explicitly that the TAO results are taken from the original publication and that preprocessing, data splits, stopping criteria, implementation details, and hardware could not be standardized. We therefore present this comparison only as supplementary evidence and do not use it to support our primary empirical claims (Section 8).
>
> **2. Runtime–accuracy trade-off and the Htru case.**
> We added practical guidance in Section 5.7. The revision clarifies that MHABR is most appropriate when a high-quality tree at a prescribed interpretable depth is important and additional offline training time is acceptable. We recommend first training CART, then applying a light MHABR configuration only when CART is insufficient, and using a more intensive configuration only when the validation improvement justifies the additional cost. We explicitly contrast Avila, where the gain is substantial, with Htru, where CART or DPDT provides a better accuracy–runtime trade-off. We also explain in Section 5.2 that runtime depends strongly on the number of distinct split candidates, reduction, and early termination, rather than only on sample size.
>
> **3. XOR intuition.**
> We moved the essential XOR intuition into the main text in Section 4.4. The new discussion explains why a one-step greedy split can show no immediate loss reduction on XOR, whereas depth-2 lookahead identifies the useful feature interaction. The complete derivation remains in Section 9.3, to which the main text now provides a forward reference.
>
> **4. Class imbalance and multiclass settings.**
> Section 5.7 now explains that MHABR applies to both binary and multiclass problems because split search is independent of the number of classes and each leaf selects the class minimizing its loss. We cite multiclass examples from our experiments and discuss the limitation of unweighted 0–1 loss on imbalanced datasets. We also describe a class-weighted extension and note that problems with many classes may require deeper trees.
>
> **5. Unified notation table.**
> We added Table 7 in the appendix, summarizing the dataset, tree, branch-and-reduce, approximation, and moving-horizon notation.
>
> **6. Interpretation of Figure 3.**
> We expanded the Figure 3 caption to summarize the main runtime findings. It now states that CART and DPDT are fastest, that Quant-BnB and DL8.5 grow sharply with depth, and that MHABR is slower than heuristic baselines but scales more moderately than the global optimization baselines.
>
> **7. Broader impact.**
> We added a broader-impact paragraph in Section 5.7. It highlights the transparency and auditability benefits of decision trees while emphasizing that interpretability alone does not guarantee fairness, robustness, or safe use, particularly in sensitive applications.
>
>
> We thank the reviewer again for the suggestions, which improved the manuscript's clarity, practical interpretation, and completeness.

---

> ### Comment · Reviewer_CTdj · 2026-07-20
> **Follow-up Discussion**
>
> I thank the authors for the reply. Sorry for my delay in engagement. I have read other responses, including the ones to the issues/questions I have raised. I am satisfied with the response.

---

> > ### Author Response · Authors · 2026-07-22
> >
> > We are grateful for the reviewer’s careful consideration and for confirming that our responses have addressed their concerns. We greatly appreciate the reviewer’s time and constructive feedback throughout the review process.

---

### Review · Reviewer_bmUi · 2026-06-26

**Summary Of Contributions:**

This paper takes on the problem of better training algorithms for decision tree classifiers, reaching a better tradeoff between optimality and scalability. Existing algorithms seem to either:

* scale well but not solve optimally. CART and others fit in this catgory

* solve (close-to) optimally but so inefficiently that they are not feasible for problems beyond depth 2 or 4 on large datasets. Methods like Quant-BnB are in this category.

This paper presents a moving-horizon approximate branch-and-reduce (MHABR) algorithm to improve the optimality-scalability frontier.

The key new insights are:

* an bilevel optimization problem (Eq 4), where the upper problem (first few levels of the tree) is solved exactly using a branch-and-reduce strategy, while the lower problem (rest of tree) is solved *approximately* via CART.

* a *moving-horizon* approach (see Sec. 4.3) that iteratively refines subtrees, closing the gap between the approximate speedup and the ideal.

NB: They could solve the both levels of their bilevel problem exactly, but this would just be too expensive.

**Additional Comments:**

First parag of Sec 4: Can you clarify in a brief way *why* the runtime of BR would be exponential, and in what variables (depth? dataset size?). Being more explicit would help the reader.

In 4.1: the "loss" of CART is not explicitly defined, just said to be symbol W. I think a clearer definition of the CART loss (even in supplement) would help

**Audience:**

Yes

**Audience Explanation:**

Yes, decision tree classifiers are an elementary method used throughout supervised machine learning. Better algorithms for decision trees would interest many TMLR readers. This is a clear Yes.

**Broader Impact Concerns:**

Not applicable. I have no concerns here.

**Claims And Evidence:**

No

**Claims Explanation:**

## Methods

I think the high level algorithms (both exact in Alg 1 and approximate in Alg 2-3) have reasonably sound design and are likely correct.

However, I had trouble following the detailed logic for reproducing some key procedures and proving several key claims in the paper, as detailed in the sections below.
I imagine that a well-execute revision could totally resolve the key questions below.


### Questions about Alg 1.

One claim that was tough to verify all the way through was the correctness of the overall procedures in Alg. 1. I have a number of questions, listed below

q1) is the value of "a" (feature index used for splitting the root) known throughout the "BR" procedure in lines 7-25? Or does the value of a change depending on the selected set from \mathbb{K}?

q2) in line 13, do you mean the *midpoint* of the set \mathcal{K}_i? or can any point be selected arbitrarily?

q3) in line 5: where does the tree "T" on the right hand side come from? do we get an initial tree from CART in line 2? this would match line 2 later in Alg 3

q4) line 17: what happens in the "update T" subprocedure? do we just point the current "root" node in the input T at the two children T_L and T_R constructed in line 14?

q5) lines 23-24: the connection between reduction and branching isn't clear from the written algorithm. can you rewrite to more clearly highlight how the \delta value is used to prune the set of possible split thresholds \mathbb{K}? Would be ideal if the pseudocode can stand alone as a high-level description.


### Questions about Lemma 4.1

Another claim that was tough to verify was the proof of Lemma 4.1

a. The three lines in Eq 18 were not clear to me... \hat{T}_L and \hat{T}_R were not defined. Plus, the definition of \hat{V} in terms of W was already provided in Eq 17 as (I think) a definition of \hat{V}, so not sure why a different path to the same result is given here in 18.

b. Below Eq 19, the equation definition a', k' relies on \mathcal{N}_c notation which I couldn't find a definition for anywhere.


### Questions about Theorem 4.2

0. Can we include the number of features as a variable in the the runtime analysis? This seems useful to quantify as specifically linear (I think?) in $p$.

1. First paragraph of the proof says the cost of CART is *linear* in the number of split candidates, which in turn means linear in the dataset size. But are you assuming sorting already happened? If so, please clarify. Otherwise there should be some O(n * log(n)) term as well, I think, given the "sort" procedure used in your Alg 5 on page 24.

2. In the second paragraph of the proof, the runtime of "BR" is examined, which I take to mean Alg 1. Again, Alg 1 does some sorting in line 4, which does not seem to be accounted for in the runtime here. Even without sorting, it is not clear to me the cost at each "layer" is O(\tilde{n}) as stated. Is the key argument that the while loop inside BnR can consider each datapoint at most once at each depth? There are some vague steps in Alg 1 ("update T", "obtain \mathcal{K}") whose runtime is tough to verify.

3. In Eq 23, is the first \sum supposed to have a start index of i=1 and stop of 2? or start index of 2 and stop of 3? The current sum limits (2 and 2) seems odd and inconsistent with the next line.

4. In Eq 23 first line, I think you want the last term's stop index to be "2^{d-1} - 1" not "2^{d-1}", right? look at Fig 2, at each depth the last node number is a power of 2 subtract one, like 7 or 15

## Experiments

The experiments in Sec. 5 are broadly sufficient for evaluating the main claims, which are whether the new algorithms here deliver better performance (primarily in terms of training error) compared to alternatives, for both the exact case (d=2) and the approximate case (d > 2).

I appreciated the number of datasets, the different scales (especially looking at "medium" and "large sizes"), and the rigor of comparison to several strong baselines.

For me, the exciting thing about the experiments was seeing the wins on the Avila dataset in Table 2: training accuracy jumps from 76% with CART to 93% with this method at depth 8, while no other method scored beyond high 70s.

Two worrisome thing about the experiments:

Worry 1: this method requires several hours on datasets of 5 million + records

* 17632 seconds (almost 5 hours) on SUSY, compared to 30 seconds by CART
* 11800 seconds (over 3 hours) on HIGGS, compared to a minute by CART

I also noticed that HIGGS is much bigger than SUSY (11 million vs 5 million, 28 features instead of 18), so any reason SUSY would take more time? Perhaps the reduce op is somehow more effective on HIGGS?


Worry 2: Are timings noisy somehow? MHABR on SUSY took 17632 sec with d=4 but only 11953 sec with d=8? How does your method get *faster* with more depth?

**Requested Changes:**

1. Revise Alg 1 to address questions above.

2. Revise proof of Lemmas and Theorems to address questions above.

3. I'd suggest renaming the two procedures in Alg 1 to something other than "BR" and "BnR" so that the difference between the two is more obvious and memorable in the name, and there's better symmetry with the corresponding approximate procedures in Alg 2/3.

4. Comment on the accuracy of timings in Table 2... why is MHABR on SUSY *faster* with depth d=8 than with d=4


*Consider as optional changes to further improve clarity*: Adjust position of Alg 3, which appears 3 pages later than its first mention on page 7.

---

> ### Author Response · Authors · 2026-07-03
>
> We thank the reviewer for the careful and constructive feedback. In response, we have substantially revised the manuscript to improve reproducibility, clarify the theoretical guarantees, and better explain the runtime results. Our detailed responses are provided below.
> ## Algorithm 1
>
> We rewrote Algorithm 1 and renamed its two procedures as `ExactTreeSearch` and `ExactSplitSearch` to make their roles easier to distinguish.
> 1. The feature index $a$ is fixed throughout each call to `ExactSplitSearch`; the outer procedure enumerates $a\in[p]$.
> 2. The evaluated split $\bar{k}$ is now explicitly defined as the midpoint of the current candidate set $\mathcal K_i$.
> 3. The incumbent tree $T$ and upper bound $U$ are initialized by CART.
> 4. The tree update is now written explicitly as  $T\leftarrow[a,\bar{k},T_L,T_R]$.
> 5. The reduction and branching steps are separated. After evaluating $\bar{k}$, we compute $\delta(\bar{k})=V_d(\mathcal I,a,\bar{k})-U$, remove candidates satisfying $|\zeta(k)-\zeta(\bar{k})|\leq\delta(\bar{k})$, and branch only on the remaining left and right candidate subsets.
> We also revised the surrounding explanation so that the pseudocode can be read independently as a high-level description of the exact search.
>
> ## Lemma 4.1
> We completely rewrote Lemma 4.1 and its proof. The undefined trees $\widehat T_L$ and $\widehat T_R$ and the undefined notation $\mathcal N_c$ were removed. The proof now uses the CART root split $(a',k')$ directly and shows $\widehat V_d(\mathcal I)\leq \widehat V_d(\mathcal I,a',k')=W_d(\mathcal I)$.
>
> We also clarified the scope of the reduction guarantee. For $d=2$, the child problems have depth 1 and are solved exactly by CART, so the exact Lipschitz bound and safe-reduction result apply. For $d>2$, reduction in ABR is presented only as a heuristic acceleration; we no longer claim that it preserves the minimizer of the approximate objective.
>
> ## Theorem 4.2 (Theorem 4.4 in the revised version)
> The complexity theorem and proof were fully revised. The number of features $p$ is now included, and repeated sorting is explicitly accounted for by an $\mathcal O(pn_t\log n_t)$ cost whenever a tree or subtree problem with $n_t$ samples is processed. The resulting bounds are
> - CART: $\mathcal O(pnd\log n)$;
> - BR: $\mathcal O(pn\log n\,\widetilde n^{d-1})$;
> - MHABR: $\mathcal O\!\left(\widetilde n pn\log n\,d(d-1)/2\right)$.
> We also replaced the previous node-index summation with the disjoint-node inequality
> $\sum_j n_j\log n_j\leq n\log n$, which removes the earlier indexing ambiguity. The revised discussion now explicitly states that BR is exponential in depth because up to $\widetilde n$ candidate root splits can recursively induce two exact child-tree problems at each depth (as a reply to additional comments 1).
>
> ## CART Loss
> We added an explicit recursive definition of the CART loss $W_d(\mathcal I)$ in the appendix and referenced it from Section 4.1. This definition specifies the greedy root split and the recursively constructed child subtrees (as a reply to additional comments 2).
>
> ## Experiments
> 1. We agree that the several-hour runtimes on SUSY and HIGGS are a practical limitation. MHABR is intended for offline settings where improved tree quality justifies additional computation; when CART already provides sufficient accuracy, CART is preferable. We have clarified this trade-off in Section 5.7 of the revised manuscript.
>
> 	We also investigated why SUSY is slower than HIGGS. Across all the runs, SUSY generated an average of 21.10 million  (3750000 for training, p=18) candidate feature-threshold pairs, whereas HIGGS generated only 8.11 million (8250000 for training, p=28), 61.6% fewer. Their repeated-value fractions were 68.74% and 96.49%, respectively. Thus, despite its larger sample size and feature count, HIGGS has a substantially smaller effective threshold-search space, which explains its shorter runtime. We have clarified this point at the end of Section 5.2 of the revised manuscript and noted that a similar nonmonotonic relationship between dataset size and runtime has also been reported in the Quant-BnB study.
>
> 2. We apologize for the reporting error. The depth-8 runtime for SUSY was incorrectly transcribed in the original manuscript. We reran the experiment using the same configuration and computing environment and updated the reported results. In the revised table, the SUSY runtime increases from depth $4$ to depth $8$, thereby resolving the inconsistency.
>
> ## Organization
> We moved Algorithm 3 closer to its first discussion, added a direct explanation of the exponential cost of exact recursive BR at the start of Section 4, and revised the notation and cross-references throughout the section.
>
> We appreciate the reviewer’s comments, which substantially improved the manuscript.

---

> > ### Comment · Reviewer_bmUi · 2026-07-17
> > **Thanks for your thorough revisions**
> >
> > Thanks for your response! I appreciate the thorough effort to address issues.
> >
> > I skimmed Alg 1 and it does look substantially better.
> >
> > I'll look at this carefully over the weekend, will plan to have an updated response by Monday.

---

> > > ### Author Response · Authors · 2026-07-17
> > >
> > > Thank you so much for the update and for taking the time over the weekend to review our revisions.
> > >
> > > We greatly appreciate your careful consideration and the considerable effort you have devoted to helping us improve the paper.
> > >
> > > We look forward to your updated feedback.

---

> > > > ### Comment · Reviewer_bmUi · 2026-07-20
> > > > **Overall my concerns are addressed. Here are some detailed comments to improve the next (hopefully final) version**
> > > >
> > > > Dear authors,
> > > >
> > > > After a close read, I'm happy to report that my overall concerns about correctness/clarity are resolved. I think the revised versions of the Algorithms and the proofs of various Lemmas and Theorems have the right details that let readers follow the story and verify the claims. The updated Experiments make sense (thanks for resolving the typo in transcribing the timing on SUSY between depth 4 and depth 8).
> > > >
> > > > Here are a few final comments that might help you revise a final version.
> > > >
> > > > ### Page 3
> > > >
> > > > if $\mathcal{K}^a$ is defined as the "index set of distinct values" of feature a, it is confusing that there are said to be n_a + 1 elements in this set, when n_a is defined as the "number of unique values". I think it is because you want \mathcal{K}^a to be the set of unique *thresholds* (ways to divide the n_a points into left/right subsets), not unique values... Please revise to say this more clearly.
> > > >
> > > > ### Eq 13
> > > >
> > > > the jump from the second line to third line seems underexplained, but maybe I'm missing something. In second line, you've regrouped to have two differences of loss functions. Focusing on the first difference in curly braces, to invoke Lemma 3.1 requires taking the difference between the size of set 1 and the size of set 2. But isn't that just what's written in third line? It seems the second difference of losses (second curly brace term) has just been skipped entirely? Maybe that's because it should be a negative and you are upper bounding with zero?
> > > >
> > > > ### Eq 16
> > > >
> > > > do you need the "> 0" at the end? the contradiction between eq 14 and 15 is in whether the V_d difference on the left hand side is a lower or upper bound of the V_d - U difference on right hand side.
> > > >
> > > > ### Lemma 4.1
> > > >
> > > > There are 3 numbered statements being proven here. but is there a difference between statements 1 and 2? Statement 1 seems like a more general case (with or without reduction) that includes 2 as a special case (without reduction). Both seem to hold for all d > 1, though only statement 2 says this explicitly.
> > > >
> > > > ### Theorem 4.4, BR subsection
> > > >
> > > >  what is lowercase $c$ in the "applying the induction hypothesis" equation? this is not defined. I'm not sure it is necessary.

---

> > > > > ### Author Response · Authors · 2026-07-22
> > > > > **Response to the Second-Round Comments**
> > > > >
> > > > > Dear Reviewer,
> > > > >
> > > > > We appreciate your careful reading and your confirmation that the revised algorithms, proofs, and experiments have addressed the main concerns. We also appreciate your helpful final suggestions. We have revised the manuscript accordingly, as detailed below.
> > > > >
> > > > > **Page 3.** We agree that the previous definition of $\mathcal{K}^a$ was confusing. We now clarify that it indexes the $n_a+1$ candidate partitions induced by the $n_a$ distinct values of feature $a$. We also revised the threshold, $\zeta(k)$, and child-set definitions to ensure that the left and right partitions are disjoint and that the notation is used consistently throughout the paper.
> > > > >
> > > > > **Equation (13).**  The second loss difference was previously handled implicitly. We have expanded this step and now apply Lemma 3.1 separately to both loss differences. Their nonnegativity is then used to derive both the upper and lower bounds, yielding the absolute-difference result directly.
> > > > >
> > > > > **Equation (16).** We agree that the final “$>0$” was unnecessary. We removed it and now obtain the contradiction directly by showing that the absolute difference between the two objective values is strictly greater than $V_d(\mathcal{I},a,\bar{k})-U$, contradicting the preceding bound.
> > > > >
> > > > > **Lemma 4.1.** We revised the statements to clarify their distinction. The first statement now gives the general bound $\widehat V_d(\mathcal{I})\leq U_{\mathrm{ABR}}(\mathcal{I},d)\leq W_d(\mathcal{I})$, regardless of whether reduction is enabled. The second statement gives the stronger equality $U_{\mathrm{ABR}}(\mathcal{I},d)=\widehat V_d(\mathcal{I})$ when reduction is disabled and all candidate root splits are evaluated. The proof has been reorganized accordingly.
> > > > >
> > > > > **Theorem 4.4, BR subsection.** The lowercase $c$ was an unnecessary implicit constant and was not defined. We have removed it and rewritten the induction step entirely using $\mathcal{O}(\cdot)$ notation.
> > > > >
> > > > > Thank you again for these comments, which helped us improve the consistency of the notation and the transparency of the proofs.

---

### Comment · Action_Editor_h9ve · 2026-06-26
**Let the discussion phase begin!**

Hi everyone!

We now have all three reviews for this paper. I want to remind you, as per TMLRs review procedure, that **we are given two weeks for the discussion phase**. This includes iterations of author responses and clarifications, possible revisions to the paper (including experiments), and reviewer feedback. For this to be a successful discussion period, I invite both authors and reviewers to be engaged and generous.

As a note to all reviewers, I will not consider your final recommendations unless I see good faith efforts to engage with the authors during this discussion period. We owe them and their work this respect.

Best,
Taylor

---

> ### Comment · Action_Editor_h9ve · 2026-07-10
> **Please engage with the author's responses**
>
> Hi all reviewers! Please engage with the authors as they have provided great responses to your reviews.
>
> It really helps our processes the sooner you are able to do this as we seek to help the authors improve their work for potential publication (as we jointly determine).
>
> Best,
> Taylor